# Brassicaceae Mustards: Phytochemical Constituents, Pharmacological Effects, and Mechanisms of Action against Human Disease

**DOI:** 10.3390/ijms25169039

**Published:** 2024-08-20

**Authors:** Mahmudur Rahman, Amina Khatun, Lei Liu, Bronwyn J. Barkla

**Affiliations:** 1Southern Cross Analytical Services, Southern Cross University, Lismore, NSW 2480, Australia; mahmudur.rahman@scu.edu.au (M.R.); amina.khatun@scu.edu.au (A.K.); 2Faculty of Science and Engineering, Southern Cross University, Lismore, NSW 2480, Australia; ben.liu@scu.edu.au

**Keywords:** Brassicaceae, bioactive constituents, canola, mustard, glucosinolates, pharmacological activity, traditional medicine, *Brassica*, *Calepina*, *Erysimum*, *Sinapis* and *Sisymbrium* species

## Abstract

The Brassicaceae genus consists of many economically important mustards of value for food and medicinal purposes, namely Asian mustard (*Brassica juncea*), ball mustard (*Neslia paniculata*), black mustard (*B. nigra*), garlic mustard (*Alliaria petiolata*), hedge mustard (*Sisymbrium officinale*), Asian hedge mustard (*S. orientale*), oilseed rape (*B. napus*), rapeseed (*B. rapa*), treacle mustard (*Erysimum repandum*), smooth mustard (*S. erysimoides*), white ball mustard (*Calepina irregularis*), white mustard (*Sinapis alba*), and Canola. Some of these are commercially cultivated as oilseeds to meet the global demand for a healthy plant-derived oil, high in polyunsaturated fats, i.e., *B. napus* and *B. juncea*. Other species are foraged from the wild where they grow on roadsides and as a weed of arable land, i.e., *E. repandum* and *S. erysimoides*, and harvested for medicinal uses. These plants contain a diverse range of bioactive natural products including sulfur-containing glucosinolates and other potentially valuable compounds, namely omega-3-fatty acids, terpenoids, phenylpropanoids, flavonoids, tannins, S-methyl cysteine sulfoxide, and trace-elements. Various parts of these plants and many of the molecules that are produced throughout the plant have been used in traditional medicines and more recently in the mainstream pharmaceutical and food industries. This study relates the uses of mustards in traditional medicines with their bioactive molecules and possible mechanisms of action and provides an overview of the current knowledge of Brassicaceae oilseeds and mustards, their phytochemicals, and their biological activities.

## 1. Introduction

Mustards are commonly cultivated worldwide and valued for their oil content in their seeds as well as for the chemical compounds they produce [1]. They are members of the Brassicaceae (syn. Cruciferae) family, commonly known as ‘mustard family’ (Table 1), and have been consumed for centuries as vegetables, and their seeds are used as a spice in condiments and as edible and industrial oils [2]. Rapeseed mustard is one of the highest oil-yielding and high protein-containing crop species, supplying around 20% of world oil production and 12 to 15% of world protein meal production [3,4]. In this study, the species of Brassicaceae that are valued for their oil content in the seed are included, other relevant species were included if they showed similar chemical constituents [1].

Mustards of the Brassicaceae family originated from the Mediterranean to the Middle East with a secondary center of origin in Asia [47]. Among the family members, *Brassica nigra* was believed to be introduced into Egypt from Asia by Aesculapius, the god of medicine, and Ceres the deity of seeds as both a potential source of food and as medicine [15]. In ancient Greece, the leaves of the young plants were eaten as spinach and the seeds were used as spices and in medicines [48]. Documentation shows that the Greek physicians including Aesculapius and scientists including Pythagoras and Hippocrates exploited the seeds for their healing powers [15,24]. Oceania, comprising the countries of Australia, New Zealand, New Guinea, Micronesia, Melanesia, and Polynesia have one of the oldest living cultural histories in the world, which goes back 50,000–65,000 years. Proof of the consumption of mustards is documented in early literature of this region [1,49,50,51,52,53,54]. Evidence of the antimicrobial effectiveness of mustard were addressed for the first time in the 1880s [55].

“Canola” is the second most important edible oilseed crop globally, second only to soybean [56]. The term Canola includes varieties of *Brassica napus* and *Brassica rapa* which are shown to have seed oil with less than 2% erucic acid and low aliphatic glucosinolate (less than 30 µmol/g) developed through selective breeding [1,57]. It is the major oilseed crop in Australia, Brazil, Canada, China, India, Europe, and the United States [58,59]. It contains low pungency, low erucic acid (less than 2% in the USA and 5% in the EU by weight) [60], and low aliphatic glucosinolate (less than 30 µmol/g) in the defatted meal [61,62]. Mustard oil is another high-value oilseed product from this family [11], popular in many Asian countries, which adds a typical hot and spicy flavor to food [16]. Traditional mustard oil or rapeseed oil (colza oil, ravison oil, sarson oil, toria oil, or turnip rape oil) is produced from the seeds of *B. napus*, *B. juncea*, *Brassica rapa* var. *campestris*, and *Brassica tourneforti* [63].

With the advent of new “omics” technologies focusing on the detection of genes (genomics), mRNA transcripts (transcriptomics), proteins (proteomics), and metabolites (metabolomics), new insights on the constituents of rapeseed and mustard, along with bioactivity data, have been reported [64,65,66,67,68,69]. The genome of *B. juncea* var. ‘*tumida*’ was first assembled in 2016 [70]. We reported shotgun proteomic analysis for *B. rapa* seeds for the first time which identified and catalogued 323 seed proteins [57,71]. Combining this proteomic data with other genomic datasets developed at Southern Cross University, a new reference genome was reported for *Brassica rapa* subsp. *trilocularis* R-o-18 and is deposited in the NCBI GenBank data base (accession GCA_017639395.1) and is being processed to appear in the EMBL-EBI Ensembl Plant Genome portal [72]. In this review, current knowledge of Brassicaceae mustards, their phytochemicals, and the biological activities of these compounds is discussed.

## 2. Phytochemical Constituents of Mustards

Mustard seeds are composed mostly of oil (28–42%), followed by protein (25–40%), carbohydrate (15–35%), fiber (10–15%), minerals (5–10%), and plant secondary metabolites (up to 10%), including glucosinolates, phenolic compounds, and tannins [59,73,74]. Plants accumulate secondary metabolites as a tolerance response to biotic and abiotic stress [75,76]. Mustard secondary metabolites are exploited for therapeutic purposes for their wide range of bioactivities [77,78,79,80,81].

### 2.1. Oil and Fatty Acids

Mustard seed oil is composed mainly of lipids in the form of triacylglycerides (Table 2), including the three major fatty acid types: saturated fatty acids (SFA; around 7–12%), monounsaturated fatty acids (MUFA; 15 to 65% of total fatty acid, depending on variety), and polyunsaturated fatty acids (PUFA; around 8–40% of total fatty acid, depending on variety) [18,22,82,83,84,85,86,87]. MUFAs consist mainly of erucic acid (C22:1, 40–50%) and oleic acid (C18:1, 5–12%). PUFAs are primarily linoleic acid (C18:2, 10–15%) and linolenic acid (C18:3, 8–15%), along with other minor fatty acids, namely eicosenoic or gondoic acid (C20:1, 5–10%), nervonic acid (C24:1, 2–8%), palmitic acid (C16:0; 2–5%), docosahexaenoic acid (C22:6, 0.1–1%), and eicosapentaenoic acid (C20:5, 0.1–1%). Linoleic acid is the major short-chained n-6 fatty acid (omega-6 fatty acids); linolenic acid, eicosapentaenoic acid, and docosahexaenoic acid are the main long-chain n-3 polyunsaturated fatty acids (omega-3 fatty acids); and oleic acid, erucic acid, nervonic acid are the commonly occurring n−9 fatty acids (omega-9 fatty acid) [18,22,82,83,84,88]. Different species of mustard and Canola oils, including *B. napus*, *B. rapa*, and *B. juncea*, typically have less SFA, moderate to high MUFA profile, and high n-3 PUFA profile and a low n-6:n-3 PUFA ratio (2:1) [22,89,90]. This lipid profile is considered one of the best amongst vegetable oils in terms of favorable health benefits, which include lowering of blood total cholesterol and low-density cholesterol, decreasing cardiovascular risk, increased insulin sensitivity and glucose tolerance, reduced inflammation, and reduced cancer cell growth [89,91].

Erucic acid is the characteristic monounsaturated long-chain fatty acid (C22:1) of the Cruciferae family and the major acyl component of most cruciferous oilseeds [86,104]. *B. rapa* var. *campestris* seeds have the highest erucic acid levels (up to 59% of total fatty acids) [95]. The trans-isomer of this acid is recognized as brassidic acid [105]. Erucic acid was found to be toxic to both rats (>0.7 g/kg body weight per day) [86,90], and humans, following prolonged intake (>7 mg/kg body weight per day), causing the gradual accumulation of lipid droplets known as myocardial lipidosis [105,106,107]. Because of this toxicity, Canola was developed from the designed breeding of several cultivars of *B. rapa* var. *campestris*, *B. juncea*, *B. napus*, *B. rapa*, and *S. alba*, bred through traditional plant-breeding techniques [108,109] and interspecific crossing between low erucic acid containing varieties, inbreeding and selection of low erucic plants within the same variety [92]. Canola (*B. napus*) was approved by the US Food and Drug Administration (FDA) as GRAS (generally recognized as safe) in 1985 for human consumption [91], however, mustard oil with high erucic acid has industrial applications such as for the formulation of lubricants and manufacturing detergents, plasticizers, polyesters, surfactants, rubber additives, elastic gums and biofuels [86,110].

The potential therapeutic usage of other fatty acids found in mustards and Canola has not been widely tested. For instance, nervonic acid, a low abundant fatty acid in mustard, is found in the white matter of animal brains and is also an important precursor for the biosynthesis of nerve cell myelin (Table 2); and docosahexaenoic acid, an omega-3 fatty acid found in mustard, is a structural element of the cerebral cortex, retina, and skin, but neither are well studied [88,111].

Other important fats and oils in Brassica seeds are diverse membrane lipids, various signaling molecules, namely galacto-oxylipins [112,113]; sterols (100–500 mg sterols/100 g seeds) like beta-sitosterol, stigmasterol, campesterol, and avenasteriol [18,114,115]; brassicasterol (24-methyl cholest-5, 22-dien-3β-ol), a lipid compound exclusive to Brassicaceae [114,116,117,118,119]; and brassinosteroids, classified as a plant hormone, which have very similar chemical structure to animal steroidal hormones and insect ecdysteroids [120,121]. In addition, there are prostaglandins, thromboxanes, and leukotrienes, as well as fat-soluble vitamins including A, D, E (tocopherols), and K (Table 3) [22].

### 2.2. Glucosinolates

Generally, all mustards have a typical pungent flavor due to the presence of amino acid-derived, sulfur-containing glucosinolates (beta-thioglucoside-N-hydroxysulfates) [132,133]. These are chemically glycosides, formed by decarboxylation on amino acids such as tyrosine, phenylalanine, and tryptophan [134]. All glucosinolates are synthesized from amino acids and sugars and share a common parent structure, consisting of a thioglycoside with a sulfonated aldoxime group that is attached to sugar via a sulfur molecule (Table 4). A side chain derived from amino acids is attached to the aldoxime moiety, which differs in various glucosinolates (Table 4). Nearly 200 individual glucosinolates are known, varying only in their characteristic side chain (Table 4 and Table 5) [134,135], and are classified into three groups—aliphatic, aromatic, and indolic glucosinolates [136]. The most abundant glucosinolates are gluconapin, sinigrin, sinalbin, glucoraphanin, progoitrin, napoleiferin, glucobrassicanapin, and glucobrassicin [8,27,137,138]. Glucosinolates and their relative amounts vary between different Brassica species, cultivars, genotypes, and within plant tissues, being affected by environment and plant age [139,140,141]. Their combined content positively correlates with oil, erucic acid, and soil sulphur content [12,86,93].

Glucosinolates are important chemical defense components that are stored in the vacuoles of Brassica leaf, root, and seed tissue, which when activated safeguard the plant against biotic stress [142,143,144]. External stimuli including tissue disruption, and insect attack compromise cell compartmentation, causing a breakdown in the natural cell partitioning of the glucosinolate from the hydrolytic thioglucosidase enzyme-myrosinase (also known as β-thioglucoside glucohydrolase) [144]. The release of myrosinase acts to hydrolyze the glucosinolate compounds into an organic aglycon (isothiocyanate, thiocyanate, or indole), glucose, and sulfate ions, in the form of potassium bisulfate (Table 4) [145,146]. The organic aglycon may then go through an intermolecular rearrangement reaction following enzyme hydrolysis. This can produce isothiocyanate (R-N=C=S) or spontaneously rearrange to give various products including nitrile (R-C≡N) and sulfur, thiocyanate (R-S-C≡N) or an oxazolidine-thione [134,147,148].

**Table 4 ijms-25-09039-t004:** The general structure of glucosinolates, their intermediate and final degradation products, and their classification depend upon the structure of their parent amino acid precursors. R is the variable amino acid-derived side chain. Adapted from [46,135,149,150,151,152,153,154].

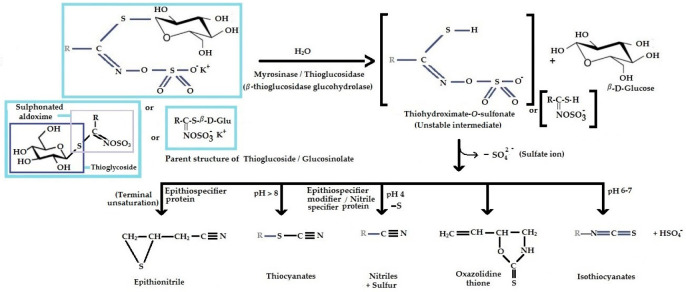			
Glucosinolate	In the Parent Structure of GSL Above, R =	Breakdown Product(s)	Structure of Breakdown Product(s)
**Aliphatic GSL (Alkenyls derived from methionine)**			
Glucocapparin	CH_3_-	Methyl ITC.	**CH_3_—N=C=S** (Methyl ITC),
Sinigrin (Allyl-GSL or 2-propenyl-GSL or potassium myronate, modern name sinigroside, C_10_H_16_KNO_9_S_2_)	CH_2_=CH-CH_2_—	Allyl ITC (2-Propenyl-ITC), allyl cyanide, 1-cyano-2,3-epithiopropane, allyl thiocyanate and allyl nitrile. After further breakdown, allyl-ITC may produce highly toxic nitriles.	CH_2_=CH-CH_2_**—N=C=S** (Allyl ITC), KHSO_4_ (Potassium bisulphate), CH_2_=CH-CH_2_-**C**≡**N** (allyl cyanide), CH_2_=CH-CH_2_**—S—C≡N** (allyl thiocyanate), CH_2_=CH-CH_2_**—C≡N** (allyl nitrile), 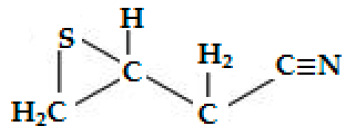 1-cyano-2,3-epithio propane
Gluconapin (But-3-enyl GSL, C_11_H_19_NO_9_S_2_)	CH_2_=CH(CH_2_)_2_- (but-3-enyl or 3-butenyl)	Butyl ITC or 3-butenyl-ITC.	CH_2_=CH(CH_2_)_2_**—N=C=S** (Butyl ITC or 3-butenyl-ITC)
Glucobrassicanapin (pent-4-enyl GSL, C_12_H_21_NO_9_S_2_)	CH_2_=CH(CH_2_)_3_- (pent-4-enyl)	Pent-4-enyl ITC or 4-pentenyl ITC.	CH_2_=CH(CH_2_)_3_**—N=C=S** (Pent-4-enyl ITC or 4-pentenyl ITC)
Glucoiberverin, (3-methyl thio propyl GSL, C_11_H_21_NO_9_S_3_)	CH_3_-S-(CH_2)3_- [or CH_3_-S-CH_2_-CH_2_-CH_2_-]	Iberverin [(3-(methyl thio) propyl ITC, (C_5_H_9_NS_2_)] and iberverin nitrile [4-(methylthio) butane nitrile].	CH_3_-S-CH_2_-CH_2_-CH_2_**—N=C=S** (Iberverin)
Glucoerucin, (4-methyl thio butyl GSL, C_12_H_23_NO_9_S_3_)	CH_3_-S-(CH_2)4_- [or CH_3_-S-CH_2_-CH_2_-CH_2_-CH_2_-]	Erucin [4-(methyl thio) butyl ITC or [isothio cyanato-4-(methyl thio)-butane]	CH_3_-S-CH_2_-CH_2_-CH_2_-CH_2_**—N=C=S** (Erucin)
Glucoberteroin (5-Methyl-thio-pentyl-GSL, C_13_H_25_NO_9_S_3_)	CH_3_-S-(CH_2_)_5_-	5-Methyl-thio-pentyl-ITC.	CH_3_-S-(CH_2_)_5_**—N=C=S** (5-Methyl-thio-pentyl-ITC)
Glucoiberin (3-(methylsulfinyl) propyl GSL, C_11_H_21_NO_10_S_3_)	CH_3_-SO-(CH_2_)_3_—	Iberin (3-Methyl sulfinyl propyl ITC) or (1-isothiocyanato-3-(methylsulfinyl)-propane) which is the Structural analog of 4-(methyl sulfinyl) butyl isothiocyanate (sulforaphane).	CH_3_-SO-(CH_2_)_3_**—N=C=S** (Iberin)
Glucocheirolin (3-methyl-sulfonyl-propyl-GSL, C_11_H_21_NO_11_S_3_)	CH_3_-SO_2_-(CH_2_)_3_-	Cheirolin [3-(methyl sulfonyl) propyl ITC] or 1-ITC-3- (methyl sulphonyl) propane.	CH_3_-SO_2_-(CH_2_)_3_**—N=C=S** (Cheirolin)
Glucoraphanin (4-methyl sulfinyl butyl GSL, C_12_H_23_NO_10_S_3_)	CH_3_-SO-(CH_2_)_4_-	Sulforaphane [1-isothiocyanato-(4R)-(methyl sulfinyl) butane] or 4-methyl-sulfinyl-butyl-ITC.	CH_3_-SO-(CH_2_)_4_**—N=C=S** (Sulforaphane)
Glucoalyssin (5-methyl-sulfinyl-pentyl-GSL, C_13_H_25_NO_10_S_3_)	CH_3_-SO-(CH_2_)_5_-	5-methyl-sulfinyl-pentyl ITC.	CH_3_-SO-(CH_2_)_5_**—N=C=S** (5-methyl-sulfinyl-pentyl ITC)
Progoitrin (2-hydroxy 3-butenyl-GSL, C_11_H_19_NO_10_S_2_)	CH_2_=CH-CH(OH)-CH_2_-	Goitrin [(2R)-2-hydroxy-3-butenyl-ITC or 5-ethenyl-1,3-oxazolidine-2-thione (C_5_H_7_NOS)], a cyclic thio-carbamate, unstable ITC and 3-hydroxy-4,5-epithio pentane nitrile.	CH_2_=CH-CH(OH)-CH_2_**—N=C=S** 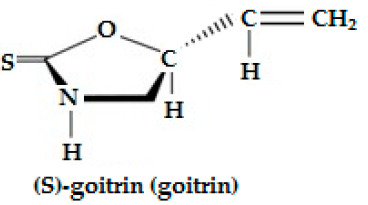
Epiprogoitrin (2(S)-hydroxy-3-butenyl GSL, C_11_H_19_NO_10_S_2_)	CH_2_=CH-CHOH-CH_2_-	Epiprogoitrin does not break down to stable ITC, a further hydrolysis produces epigoitrin [(R)-5-vinyloxazolidine-2-thione] and finally goitrin.	CH_2_=CH-CH(OH)-CH_2_**—N=C=S** 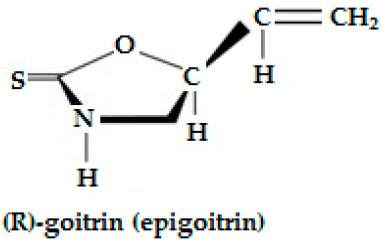
Glucoalyssin (5-methyl-sulfinyl-pentyl-GSL, C_13_H_25_NO_10_S_3_)	CH_3_-SO-(CH_2_)_5_-	5-methyl-sulfinyl-pentyl ITC.	CH_3_-SO-(CH_2_)_5_**—N=C=S** (5-methyl-sulfinyl-pentyl ITC)
Glucolepidiin (Ethyl GSL, C_9_H_17_NO_9_S_2_)	CH_3_-CH_2_-	Ethyl isothiocyanate.	CH_3_-CH_2_**—N=C=S** (Ethyl isothiocyanate).
1-cyano-2-hydroxy-3-butene (or 2-hydoxybut-3-enyl cyanide, CH_2_=CH-CHOH-C**≡N**)- a **Nitrile** **(R-C≡N)**	-	3-butenonitrile (2-methyl-3-butenenitrile), 4-methylthiobutanonitrile (4-methylsulfanylbutanenitrile oxide).	CH_2_=CH-CH_2_-C**≡N** (3-butenonitrile) CH_3_-S-CH_2_-CH_2_-C**≡N=O** (methylthiobutanonitrile)
**Indolyl GSLs (Indolyl derivatives from tryptophan)**			
Glucobrassicin (3-indolyl methyl GSL, C_16_H_20_N_2_O_9_S_2_)	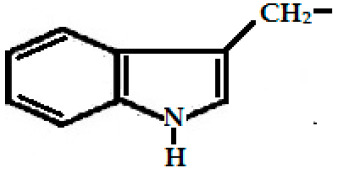 (3-indolyl methyl-)	Myrosinase breaks down glucobrassicin to 3-indoly-methyl-ITC (unstable) and simultaneously releases thiocynate anion (SCN-) to yield indole-3-carbinol (3-hydroxymethyl- indole). Indole-3-carbinole can combine with ascorbic acid to form Ascorbigen.	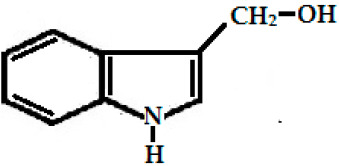 indole-3-carbinol
Napoleiferin (gluconapoleiferin) (2-hydroxy-4-pent enyl GSL, C_6_H_9_NOS); it is a natural homolog of goitrin.	R=CH_2_CH(OH)CH_2_CH=CH_2_- (2-hydroxy-4-pent enyl-)	Oxazolidine-2-thione.	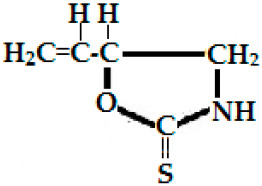 Oxazolidine-2-thione
Neoglucobrassicin (1-Methoxy-3-indolyl methyl GSL, C_17_H_22_N_2_O_10_S_2_)		Hydrolysis yields unsTable 1-methoxy-3-indolyl methyl ITC and thiocyanate ion (SCN^−^).	
**Aromatic GSL (Arylalkyls derivatives from phenylalanine)**			
Sinalbin (C_30_H_42_N_2_O_15_S_2_) (p-hydroxy benzyl GSL, sinapine glucosinalbate) or 4/p-hydroxybenzyl GSL or glucosinalbin), the choline ester of sinapic acid	p-OH-C_6_H_4_-CH_2_-	Allyl isothiocyanate, and on hydrolysis by myrosinase, produces unstable para hydroxy benzyl ITC. Para hydroxy benzyl ITC can further decomposes to para hydroxyl benzyl alcohol and thiocyanate ion (SCN^−^) and/or corresponding hydroxyl derivatives like di-(para hydroxyl benzyl disulphide). It also produces p-hydroxy benzyl amine known as the ‘white principles’.	CH_2_=CH-CH_2_**—N=C=S** (Allyl ITC), p-OH-C_6_H_4_-CH_2_**—N=C=S** (para hydroxy benzyl ITC), p-OH-C_6_H_4_-CH_2_—OH (para hydroxyl benzyl alcohol), HO—C_6_H_4_-CH_2_—NH_2_ [p-hydroxy benzyl amine (4-Hydroxybenzylamine)]
Gluconasturtiin (2-phenyl ethyl-GSL, C_15_H_21_NO_9_S_2_)	R=C_6_H_5_-CH_2_-CH_2_-	Phenethyl ITC (2-penyl-ethyl ITC).	C_6_H_5_-CH_2_-CH_2_-N=C=S (Phenethyl ITC)
Glucotropaeolin, Benzylglucosinolate (C_14_H_19_NO_9_S_2_)	R=C_6_H_5_-CH_2_-	Benzyl ITC, benzyl cyanide and benzyl thiocyanate.	C_6_H_5_-CH_2_-N=C=S (Benzyl ITC)

GSL = Glucosinolate, ITC = Isothiocyanate. Aliphatic glucosinolates are also further classified into three classes (i) three carbon chain length aliphatic glucosinolates (i.e., sinigrin, glucoiberin, and Glucoiberverin), (ii) four carbon chain length aliphatic glucosinolates (i.e., glucoerucin, dehydroerucin, glucoraphanin, glucoraphenin, gluconapin, and progoitrin), and (iii) five carbon chain length aliphatic glucosinolates (i.e., glucoberteroin, glucoalyssin, glucobrassicanapin and gluconapoleiferin) [135].

**Table 5 ijms-25-09039-t005:** Biological activities of major glucosinolates and their metabolites found in mustards.

Glucosinolates (GSL)	Mustards	Biological Activity of GSLs and Their Metabolites (Referred from Table 4)	References
**Glucosinolates (secondary metabolites)**			
Sinigrin, allyl glucosinolate (C_10_H_16_KNO_9_S_2_)	*Brassica nigra* (1% sinigrin), *Sinapis alba*, *B. rapa* var. *campestris*, *B. carinata*, *B. juncea*, *Calepina irregularis*, *Alliaria petiolata*, *B. rapa*, *B. napus*	Anti-cancer, anti-inflammatory, antibacterial, anti-fungal, antioxidant, and wound healing activity. Allyl-isothiocyanate is responsible for internal and external irritation and local vasodilation. It has been shown to prevent urinary bladder cancer and anti-proliferative activity against human prostate cancer cells. 1-cyano-2,3-epithiopropane has in vitro anti-cancer activity against human hepatocellular carcinoma.	[22,31,118,137,146,147,151,155,156,157,158,159,160,161,162,163,164]
Gluconapin, 3-Butenylglucosinolate, Butyl isothiocyanate (C_11_H_19_NO_9_S_2_)	*B. rapa* var. *campestris*, *B. juncea*, *B. napus*, *Sinapis alba*	Tumour inhibiting activity.	[22,147,160,165,166,167,168,169]
Glucocheirolin (C_11_H_20_NO_11_S_3_)	*Calepina irregularis*, *Erysimum corinthium*	Anti-proliferative activity on cancerous cells. Cheirolin improves cellular antioxidant defense and stress response mechanisms.	[31,34,146,155,170]
Glucoraphanin, 4-methylsulphinyl-butyl glucosinolate (C_12_H_23_NO_10_S_3_)	*B. rapa* var. *campestris*, *B. nigra*	Used in the treatment of neurodegenerative disorders as an antioxidant, gastric ulcers caused by Helicobacter pylori; against cancer like fibroblasts and malignant melanoma and employed to improve autism. Sulforaphane is a strong cancer chemopreventive agent both in vitro and in vivo and a potent monofunctional phase II enzyme inducer.	[22,24,30,38,155,171,172,173,174,175]
Progoitrin, (R)-2-hydroxybut-3-enylglucosinolate (C_11_H_19_NO_10_S_2_)	*Brassica napus*, *B. rapa* var. *campestris*, *B. juncea*	Thyroperoxidase enzyme inhibition and interference with the uptake and use of iodine by the thyroid gland. Anti-cancer effects.	[22,147,151,160,165,166,176]
Epiprogoitrin (C_11_H_19_NO_10_S_2_)	*Brassica napus*, *B. olarecia*	Nematicidal activity and insecticidal property as an effective fumigant.	[152,177,178]
Glucobrassicanapin, 4-pentenyl glucosinolate (C_12_H_21_NO_99_S_2_)	*B. rapa* var. *campestris*, *B. napus*, *B. rapa*	Anti-cancer activity	[22,147,165]
Napoleiferin (C_6_H_9_NOS)	*Brassica rapa*, *B. oleracea*, *B. napus*, *B. rapa* var. *campestris*, *B. nigra*, *B. juncea*, *B. carinata*.	Goiterogenic substance.	[167,179,180,181,182]
* **Nitriles R-C≡N** *			
Erucin, 1-isothiocyanato-4-(methylthio)-butayl isothiocyanate (C_6_H_11_NS_2_)	*Eruca sativa* Mill. (rocket).	Erucin is a sulfone analog of sulforaphane. It is an enzymatic hydrolysis product of glucoerucin, responsible for the characteristic aroma of broccoli. Chemically, it is the reduced form of sulforaphane. It has antioxidant, neuroprotective, anti-fungal, and anti-inflammatory activities. It has strong cancer chemopreventive activity. It induces apoptosis in cancer cells, exerts antiproliferative effects, and causes cell cycle arrest in cancer cells.	[183,184,185,186]
Erucin nitrile (1-cyano-4-(methylthio)butane) may	*Brassica oleracea* var. *italica* (broccoli)-seeds, sprouts and root	Erucin nitrile could be formed in the anaerobic environment of the cecum or by hydrolysis of glucoerucin by reduction of sulforaphane nitrile. Erucin nitrile has potential activity as a phytoalexin, i.e., antimicrobial and antioxidant activities.	[183,186]
Sulforaphane nitrile, 5-(methylsulfinyl) pentanenitrile (C_6_H_11_NOS)	*Brassica oleracea* var. *italica* (broccoli) sprouts, *Brassica oleracea* var. *botrytis* (cauliflower)	Myrosinase cofactor epithiospecifier protein breaks down glucoraphanin to sulforaphane nitrile. Both sulforaphane and sulforaphane nitrile have chemoprotective, antioxidant, and anti-cancer effects, but sulforaphane nitrile is substantially less potent than sulforaphane.	[187,188]
** *Epithionitrile* ** 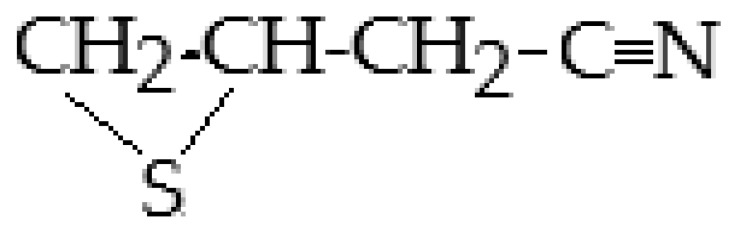			
** *Epithionitriles, major hydrolysis products * **	*Brassica rapa*, *B. oleracea*, *B. napus*, *B. rapa* var. *campestris*, *B. nigra*, *B. juncea*, *B. carinata*.	3-butenonitrile has anti-cancer and antimicrobial activities. 4-methylthiobutanonitrile has a characteristic aroma. 1-cyano-2,3-epithiopropane, and 1-cyano-3,4-epithiobutane, 1-cyano-4,5-epithiopentane have cancer-preventative activities.	[189,190,191,192]
Glucoalyssin (C_13_H_25_NO_10_S_3_)	*B. rapa* var. *campestris*, *B. napus*, *Degenia velebitica*	The flavor component of cooked Brassicas. It is an antioxidant.	[163,193]
Gluconapoleiferin (C_12_H_21_NO_10_S_2_)	*B. napus*, *B. rapa*, *B. rapa* var. *campestris*, *B. juncea*, *B. nigra*	Tasteless, but hydrolytic, susceptibility to peptic digestion. A precursor to glucobrassicanapin biosynthesis. Thyroid gland function inhibitory effect.	[163,194]
Glucolepidiin (C_9_H_17_NO_9_S_2_)	*Calepina irregularis*	Antibacterial activity.	[31,146]
** *Thiocyanates* ** ** *R-S-C≡N* **			
Glucobrassicin, the unsubstituted indole glucosinolate. (C_16_H_20_N_2_O_9_S_2_)	*B. juncea*, *B. napus*	Antioxidant, anti-inflammatory, and anti-cancer activities. It stimulates the bodily natural detoxifying enzymes.	[151,160,163,166,167,195,196,197]
Neoglucobrassicin (C_17_H_22_N_2_O_10_S_2_)	*B. napus*, *S. officinale*.	Antioxidant and anti-cancer activity.	[151,163,195]
Glucoiberin (C_11_H_21_NO_10_S_3_)	*B. napus*, *B. incana*, *B. oleracea*, *Calepina irregularis*, *Erysimum corinthium*, *Iberis amara*, *Moringa oleifera*	Anti-cancer effects	[22,31,34,146,147,151,165]
Glucoiberverin (C_11_H_21_NO_9_S_3_)	*Calepina irregularis*	Repellent and acts against insect herbivory	[31,146].
Glucoerucin (C_12_H_23_NO_9_S_3_)	*Calepina irregularis*	Inhibition of cancerous cell proliferation.	[31,146,155].
Sinalbin, P-Hydroxybenzyl Glucosinolate (C_14_H_19_NO_10_S_2_)	Sinapine is present in all the mustards in different amounts. Chiefly found in white mustard *Sinapis alba* (2.5% of seed weight) and also reported in *B. rapa* var. *campestris*, *B. napus* *B. juncea* and *B. nigra*	Antioxidant, anti-microbial, and anti-fungal activity. Sinapine and sinamic acid have antimicrobial, antioxidant, and anti-cancer properties.	[22,28,58,147,157,165,198,199,200]
** *Isothiocyanates* ** ** *R-N=C=S* **			
Glucotropaeolin or benzyl glucosinolate, precursor of benzyl isothiocyanate (C_14_H_19_NO_9_S_2_) [C_6_H_5_-CH_2_-GSL]	*Alliaria petiolate*, *B. juncea*; *B. nigra*, *Calepina irregularis*, *Lepidium sativum* (garden criss), *Sinapis alba*.	Benzyl isothiocyanate or Glucotropaeolin has potential cancer-preventive activities, anti-tumorigenic (lung and esophagus in humans) activity, and works as an antimetastatic agent. It is also used as a food preservative because of its antimicrobial activity.	[31,146,155,160,166,169,201,202,203]
Gluconasturtiin or Phenethylglucosinolate, the precursor of phenethyl isothiocyanate (C_15_H_21_NO_9_S_2_)	*B. napus*, *Nasturtium officinale* (watercress)	Inhibits the development of tobacco-specific carcinogen-induced lung tumors.	[22,147,151,165,201,204]
Sulforaphane, 1-isothiocyanato-4-(methyl- sulfinyl) butane (C_6_H_11_NOS_2_). Sulforaphane is derived from glucoraphanin.	*B. oleracea* var. *italica* (broccoli) sprouts, kale, *B. oleracea* var. *botrytis* (cauliflower) or *B. oleracea* var. *gongylodes* (kohlrabi)	Anticarcinogenic activity by induction of cell cycle arrest and apoptosis in various human cancer cells; antioxidant, antiproliferative, anti-inflammatory, antimicrobial, anti-fungal, and antiviral activities.	[79,186,205,206,207]
Indole-3-carbinol, derived from glucobrassicin. (C_9_H_9_NO)	*B. oleracea* var. *italica* (broccoli), *B. oleracea* var. *acephala* (kale), *Brassica oleracea* var. *capitata* (cabbage)	Anti-cancer, antioxidant, and antiatherogenic activities.	

### 2.3. Phenolics and Flavonoids

Hydroxyl group-containing aromatic compounds known as phenolic or polyphenolic compounds represent the most abundant plant secondary metabolites in rapeseed, almost 50% of secondary metabolites, namely phenolic choline esters, predominantly sinapate choline esters [208]. They occur mainly in the form of phenolic acids, flavonoids, and soluble and insoluble condensed tannins [209]. These compounds can be present as free or bound in the form of esters and glycosides. The most abundant free phenolic acid found in oilseed mustards is p-hydroxybenzoic acid [115], whereas sinapic acid, also known as sinapinic acid, a naturally occurring hydroxycinnamic acid, is the most abundant bound phenolic acid (90–95% of the total phenolic acids identified in rapeseed) [210], and is present mainly in the esterified form [59,210]. The choline ester of this short-chain fatty acid is sinapine, an alkaloidal amine (Figure 1), and is an integral component for the biosynthesis of lignin and flavonoids in plants [74]. Oilseeds (excluding Canola) contain 12–15 g sinapine per kilogram of seeds [28]. The other minor phenolics are apigenin (4′,5,7-trihydroxyflavone), syringic acid, 1,2-disinapoylglucose [165,211,212], p-hydroxybenzoic, p-coumaric, trans-ferulic acid, gentisic, gallic, protocatechuic, syringic and vanillic acids (Figure 1) [213,214]. These antioxidant polyphenolics act as a preservative, preventing the reserved oil in the seed from degrading over time [215]. As a consequence of this, they also act as preservatives to counter the rancidity of the extracted oil, preventing it from discoloring and becoming cloudy [216]. Studies indicate that seed roasting reduces the amount of phenolic compounds in mustard seeds and increases the oxidative stability [217,218,219].

Flavonoids are composed of a fifteen-carbon-containing skeleton where two aromatic rings are interlinked by a three-carbon chain (C6-C3-C6) [209,220]. They are found in the seeds of many mustard species and include proanthocyanidins and anthocyanidins; flavan-3-ols, namely catechin and epicatechin; flavonols including kaempferol and quercetin; and flavonoid glycosides [6,10,38,39,43,44,160,166,212,214]. Phenolic acids and tannins give the astringency and dark color to commercially processed seed extracts [59,74,210]; specifically, Kaempferol 3-O-(2‴-O-Sinapoyl-β-sophoroside) has been identified as the major compound responsible for the unpleasant bitter taste of mustard protein isolates [221]. Oriental mustard oil (*B. juncea*) is notably darker in color due to the presence of higher phenolics than yellow mustard (*S. alba*) [222].

**Figure 1 ijms-25-09039-f001:**
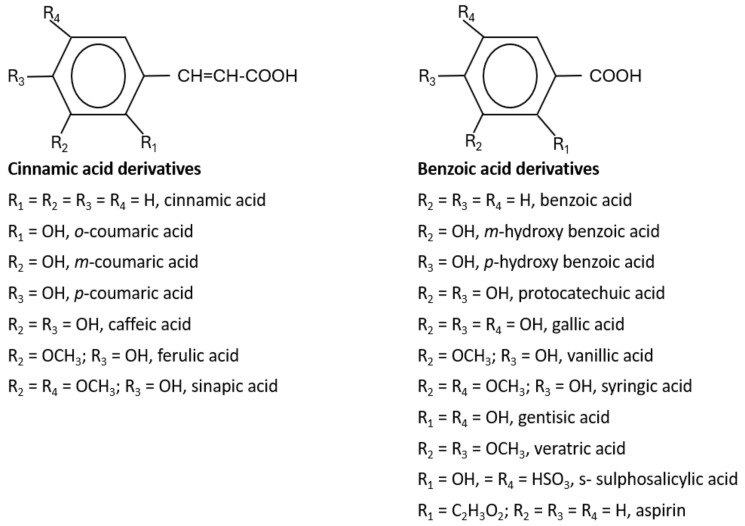
Basic structures of phenolic acids found in Brassicaceae oilseeds. Adapted from [59,209,223,224,225].

### 2.4. Proteins

Mustard seeds contain high amounts of proteins and the de-oiled seed contains up to 40% proteins [74,160,161,226]. Among them, the major seed storage proteins are cruciferin, napin, and the oil-body protein oleosin [74]. Cruciferins are 12S legumin-type globulin proteins with a molecular weight of about 300–420 kDa [227,228]. These proteins comprise up to 35–50% of the total seed protein. Napins belong to the 2S albumin seed storage protein family, make up 20–40% of total seed protein, and have a molecular weight in the range of 12–16 kDa [74,229]. Oleosins, which are the main protein found in the lipid bodies of the seeds, make up 6–8% of the total seed protein and have a molecular weight of approximately 18–25 kDa [74,229,230]. The other relatively abundant proteins found in mustard seeds are myrosinase, the glucosinolate-degrading plant defense enzyme found in all mustards which has a molecular weight of 120–150 kDa [158,159,231] and thionins, now also known as defensins, low-molecular-weight (about 5 kDa) cationic peptides abundant in sulfur-containing cysteine residues with plant defense action [232,233]. Additionally, heat-sensitive trypsin inhibitor proteins of 18–19 kDa ranging from 2–2.5% of crude protein [62,234,235] and lipid transfer proteins of 7–9 kDa which accounts for 2–3% of the total protein, are also found in the seed [236,237,238].

### 2.5. Minor Chemical Compounds

The amount of carbohydrates is comparatively lower in the seeds of rape and Canola than that of other oilseeds and seed grains [239]. In general, the seeds contain around 10% soluble sugars, 4–5% cellulose, 4–5% pectin, 3% hemicellulose, and less than 1% starch of dry seed weight [59]. The levels of oligosaccharides were found to be approximately 20 g/kg dry matter and raffinose was approximately 3–4 g/kg dry matter in seed meal [240].

High levels of the major minerals including iron (0.01–0.02%), calcium (0.3–0.45%), phosphorus (0.6–0.84%), potassium (0.5–0.65%), zinc, magnesium (0.1–0.15%), and manganese (0.003–0.008%) are found in the seed. Metals such as copper (0.001–0.007%), lead, cadmium, and selenium (0.00002%) have also been measured in the seeds [123,241,242,243,244,245,246,247]. Mustard seeds contain crude fibre including pentosans and lignin, which accounts for 7–15% of the total seed weight [28,198].

## 3. Use of Mustard to Treat Ailments

Mustards are described in many official herbal compendia including The British Pharmacopoeia, The Pharmacopoeia of China, and other publications of herbal, Ayurvedic, and homeopathic relevance as a natural ointment to heal various forms of pain [14,43,137,248,249]. They have been used for years for the treatment of disease and pathological conditions (Table 6).

The traditional uses of mustards and their components in vitro and in vivo have been well documented in the literature [272,273,274]. The rationale and possible mechanism of action behind the uses are discussed in the following sections.

### 3.1. Antimicrobial and Antiviral Activities

Glucosinolates demonstrate a wide range of antibacterial and anti-fungal properties with direct or synergistic effects in combination with other compounds. Different glucosinolate metabolites, namely nitriles, thiocyanates, and isothiocyanates, can strongly inhibit most Gram-positive and Gram-negative bacteria [275], as well as fungus and yeast [149,252]. In addition, other hydrolysis products of glucosinolates such as allyl nitrile, allyl isothiocyanate and ionic thiocyanates (SCN-), phenolic compounds, and tannins have also demonstrated potential anti-microbial activity [276,277]. The best studied of the mustard seed anti-microbial metabolites is allyl isothiocyanate along with its related isothiocyanates [278]. These hydrophobic compounds have a multi-targeted mechanism of action, affecting bacterial metabolic pathways, membrane integrity, and cell wall structure [216]. They have the ability to disrupt the bacterial cell membrane, causing the leakage of both metabolites and ions and a reduction in both intracellular ATP and pH [216,252,279,280] resulting in cell death. This mechanism of action is quite similar to the class of polymyxin and related peptide antibiotics, both of which have strong affinities for membrane lipids and lipopolysaccharide [281,282].

Natural isothiocyanates and their analogs have also been shown to be effective against a wide range of fungi including *Aspergillus niger*, *Penicillium cyclopium*, *Rhizopus ryzae*, and common *Saprophytic* fungi [283]. Allyl isothiocyanate was found to be toxic [284] to *Penicillium expansum*, a fungus which causes postharvest disease to economically important fruits like apples and vegetables in storage [285]. Benzyl isothio-cyanate was found effective against ectomycorrhizal fungi [203,286].

Sinapic acids and 4-hydroxy-3-nitrophenylacetic acid were found to have antibacterial activity against several different pathogenic bacteria. Structure–activity relationship studies revealed that different methoxyl, hydroxyl, nitro, and propenoic groups present in the compounds are necessary for effectiveness [287].

Several seed proteins have been shown to have both anti-microbial and anti-fungal activities. The seed storage protein napin was reported to have significant antimicrobial activity against pathogenic bacteria, namely *Bacillus subtilis*, *B. cereus*, *B. megaterium*, and *Pseudomonas aeruginosia*, and against the highly toxigenic fungus *Fusarium langsethiae* [281,288,289,290,291,292,293,294,295]. The anti-microbial activity of the seed storage protein napin was attributed to the presence of abundant positively charged amino acids, giving the protein a net charge of +1 or greater. A high proline content and a high proportion of alpha-helices in their secondary structure were also found to be important. It is thought that these properties enable the protein to compete for binding with calmodulin and inhibit calcium signaling. The anti-tryptic activity of napins could also hinder microorganism growth [73,292,296]. There is also evidence that the basic residues of napin create electrostatic contacts with the acidic phospholipid membrane bilayer of the pathogen and thus permeabilize the microbial plasmalemma [297,298], as well as inhibit their growth [299].

Another seed storage protein, vicilin, which was reported in *B. rapa* for the first time among the Brassicaceae mustards [281], has also been shown to possess antimicrobial activity [300]. Thionins have been reported to have potential antimicrobial activity against *Clavibacter michiganensis*, *Micrococcus luteus*, *Rhizobium meliloti* and *Xanthomonas campestris*, and toxic fungus *Fusarium solani* [236,299,301]. Noteworthy, in silico studies have found that both the napin and cruciferin seed storage proteins have high binding affinity to selective metabolic enzymes [62,281] and microbial membranes [302] in pathogenic bacteria. These findings suggest that these proteins are able to form protein–enzyme complexes that may cause alteration of regular bacterial metabolism, plasma membrane permeability, and chemical or reactive oxygen species balance as the conformation of the enzyme or membrane changes, leading to cell death [289,303,304,305].

Interestingly, applying these compounds together has shown additive antibacterial and anti-fungal effects. In the case of napins, these proteins were found to exert synergistic effects when applied in combination with thionines by interacting with membrane phospholipid permeabilization in the case of bacteria and increasing K^+^ efflux from fungal hyphae and changing membrane ion permeability [306]. Relevantly, glucosinolate hydrolysis products from the leaf, stem, and inflorescence of broccoli (*Brassica oleracea* var. *italica* L.), particularly iberin- and indole-derived glucosinolate hydrolysis products, which are also present in the Brassicaceae mustard seeds, were shown to have anti-fungal activity against *Botrytis cinerea*, a necrotrophic fungus that infects many plant species [307].

Mustard seed extracts are also recommended to fight against candidiasis, a fungal infection caused by an overgrowth of any species of the genus Candida yeast [308,309]. Mustard seed extracts are also given for the treatment of other topical fungal infections [310]. The antimicrobial and antimycotic activities of mustard seed support their uses in human diseases involving bacteria and fungi as well as the use in herbal and food preparations as preservatives [281,301,308,311,312].

Rapeseed protein hydrolysates were also tested in *E. coli* expressing the HIV protease. Measuring the inhibitory activity, it was found that the proteins act as the inhibitor of human immunodeficiency virus (HIV) protease which opens the possibility to develop antiviral drugs as well as combat HIV [313]. Brassinosteroids, a group of steroidal substances, found in most of the mustard seeds, were found to have substantial anti-viral activity against human pathogenic viruses including herpes simplex virus type 1, RNA viruses, and even the measles virus [314].

### 3.2. Antidiabetic Action

Diabetes mellitus is a major endocrine disease in which the pancreas produces insufficient insulin or the produced insulin cannot bind with the appropriate receptor. The lack of insulin signaling leads to high blood sugar levels which can cause secondary complications and in some cases be fatal. Anti-diabetic drugs, medications used to treat diabetes mellitus by decreasing glucose levels in the blood, and hypoglycemic agents, the agents used to help reduce the amount of sugar present in the blood, are used to treat this growing global health problem [16,315].

Seeds of *B. juncea* and *B. nigra* have been reported to have anti-diabetic properties, including the ability to lower blood glucose and improve high glucose tolerance in diabetic animals [253,316,317]. Mustard seeds have also been found to suppress gluconeogenic and glucolytic enzymes, including *α*-glucosidase [254,317,318]. This enzyme catabolizes the breakdown of complex carbohydrates to glucose and thus its inhibition would ultimately decrease blood glucose levels [319]. The flavonoids found in rapeseed including flavones, isoflavones, flavanones, flavonols, and anthocyanidins have the ability to reverse the signs of type II diabetes. For example, abscisic acid, a plant growth regulator, in the flavonoid biosynthesis pathway, can act in a similar manner to the widely used thiazolidinediones class of anti-diabetic drugs by targeting the thiazolidinedione receptor and enhancing insulin secretion [320].

*B. juncea* seed powder extract has also been shown to have significant hypoglycemic action [254,317], increasing the concentration of hepatic glycogen by increasing glycogenesis via glycogen synthetase activity. It also increases glycogen by decreasing glycogenolysis and gluconeogenesis via the depressing activity of glycogen phosphorylase and gluconeogenic enzymes [253].

The increased blood glucose levels in diabetes mellitus generate various free radicals and reactive oxygen species including superoxide anions, which generate hydroxyl radicals via the Haber–Weiss reaction. This results in the peroxidation of membrane lipids and ultimately leads to oxidative damage to cell membranes [256]. Antioxidant molecules present in mustard can help to neutralize the harmful free radicals by different mechanisms including electron donation, metal ion chelation, or by gene expression regulation [215,317,318], safeguarding membrane lipids from lipid peroxidation [321,322].

It is also well known that food rich in MUFA, such as has been shown for mustard seed (Section 2.1), can increase the peripheral insulin sensitivity of the insulin receptor and thus lower blood glucose levels [323,324]. Fibers present in foods, particularly soluble fibers, can also affect the glucose and insulin levels in the blood [325]. Mustard mucilage (soluble fiber) at different dietary levels was found to have antidiabetic effects on elevated blood sugar levels (post-prandial glucose level) and insulinemia in laboratory rats [253,254,326,327]. Measuring hypoglycemic activity is the most common and accepted method to evaluate the antidiabetic activity of natural products. The complete profiling of the hypoglycemic activity of mustard components as well as human trials are scarce for antidiabetic activity and warrants further investigation. Antihyperglycemic agents reduce the excess accumulated sugar in the blood [328]. *B. juncea* oil and seeds showed potential antihyperglycemic effects, i.e., a significant reduction in blood glucose levels in experimental diabetic rats by increasing insulin activity [328,329]. Leaves of *B. juncea* and *B. compestris* exert similar antidiabetic effects and anti-hyperglycemic effects in diabetic mice [273,330].

Rapeseed protein hydrolysates derived peptide RAP-8 was tested on an in vivo mouse model to assess its effect on glucose tolerance and insulin resistance. The peptide was found to significantly improve insulin resistance, glucose intolerance, and lipid metabolism as well as markedly reduce hepatic inflammation, fibrosis, liver injury, and metabolic deterioration in mouse models of non-alcoholic steatohepatitis and hepatic fibrosis [331].

### 3.3. Hypolipidemic Effects

Hypolipidemic compounds are substances which lower the lipid and lipoprotein levels in the blood while hypocholesterolemic compounds decrease cholesterol absorption and increase bile acid excretion, thus reducing serum cholesterol by stimulating the further conversion of cholestenol to bile acids [332,333,334].

The low SFA and high MUFA profile of mustard oil (Table 2) has the potential to lower cholesterol. Studies conducted on healthy human subjects revealed that high MUFA diets reduce both blood cholesterol and triacylglycerol levels (total cholesterol by 10% and LDL cholesterol by 14%) [323]. The ingestion of rapeseed oil was found to reduce cholesterol absorption from food and increase the excretion of cholesterol, bile acids, and metabolites of steroid hormones from the digestive tract, thus reducing the serum cholesterol level in the blood [332]. Rapeseed oil was also found to reduce low-density lipoprotein (LDL) cholesterol, blood lipids, and plasma apo-lipoproteins in a study which compared the effects of consumption with olive oil in healthy male participants [335]. *B. nigra* seeds, added to the diet of diabetic rats, reduced serum cholesterol and triacylglycerol levels and raised the level of good cholesterol (high-density lipoprotein, HDL) [316]. Seeds of *B. juncea* were tested on 1–2 dimethyl hydrazine-induced colon carcinogenesis and found to decrease the plasma cholesterol and phospholipid levels while increasing the fecal bile acids and neutral sterols [336,337]. Erucic acid present in all mustards was tested and found to exhibit hypocholesterolemic and hypolipidemic effects on the plasma and tissues of hypercholesterolemic rats [333].

Another mechanism that serves to lower cholesterol is the degradation of consumed fiber by the intestinal flora which produces propionic acid, that is then absorbed into the blood stream, and serves to inhibit hepatic cholesterol synthesis [338,339]. In addition, most of the phytosterols found in mustards, namely brassicasterol, campesterol, sitosterol, avenasteriol, and stigmasterol have been shown to have plasma cholesterol-lowering activity [90,114]. A peptide from rapeseed protein hydrolysates demonstrated the ability to improve lipid metabolism in experimental mouse models [331]. Improved lipid metabolism significantly leads to the burning of stored fats, weight loss, cardiac, pulmonary, and metabolic performance; delays aging; and ultimately optimizes bodily physiological functions [340,341].

### 3.4. Cytotoxic and Anti-Cancer Activity

Mustard and its components are alleged to have cytotoxic and anti-cancer properties mainly due to their demonstrated anti-oxidative properties [342]. Different isothiocyanate, erucic acid, phenols, and phytins possess the ability to scavenge free radicals which may have anti-cancer effects [14,343]. Erucic acid was found to possess anti-tumor effects on animal models as well as human cancer cell lines [344]. Sinigrin, progoitrin, and glucocheirolin showed a marked in vitro cytotoxicity against human erythroleukaemic K562 cells [152]. Mustard oilseeds also demonstrate higher antioxidant potential in terms of DPPH (2,2-diphenyl-1-picrylhydrazyl) scavenging potential, ferric-reducing ability, total phenolic content, and chelating power [345,346]. It was found that the consecutive hydrolysis of rapeseed proteins using the digestive enzymes pepsin and pancreatin exhibited the most active DPPH radical scavenging hydrolysate [347].

Antioxidant molecules act as free radical scavengers; they neutralize free radicals produced from the body cells by delivering some of their own electrons to the free radicals produced from the normal essential bodily cell metabolic processes or from external sources. Mustard oilseeds are a great source of antioxidants which scavenge free radicals including phenolic compounds, ligands, flavonoids, and phenolic acids (Table 5, Figure 1) [218]. Antioxidant molecules target cellular free radicals, neutralizing the excess reactive oxygen species, and may ultimately help to reduce the potential risks of developing cancer [348,349,350].

More direct evidence for the anti-cancer effects of mustard compounds has been shown in studies using cancer cell lines and animal models. Sinigrin and its metabolites isolated from the leaves of *B. carinata* were reported to have potential tumor growth inhibition activity in studies on both in vivo animal models and in vitro cell-line models [12]. Extracts from *B. juncea* seeds were shown to decrease the cancerous cells in the colon and intestine in 1–2 dimethyl hydrazine-induced colon carcinogenesis in rat models [336,337]. Recent research has demonstrated that the mucilage/meal fraction obtained from *Sinapis alba* following oil extraction contains a complex mixture of polysaccharides which were shown to exert a protective role in the development of sporadic and obesity-associated colon cancer in preclinical azoxymethane-induced obese rat models compared to untreated obese rats [351].

Mustard seed extract was also reported to act by suppressing the expression of lymphocyte activating factor, tumor necrosis factor alpha (TNF), and interleukin (IL)-6 mRNA, and inhibiting Langerhans cell migration in the epidermis of Albion mice cancer models [352]. More specifically, a regulatory protein NPR1, found in *B. juncea* seeds, has been shown to inhibit the activity of nuclear factor-kappa B (NF-κB), a member of the family of dimeric transcription factors which regulate immune responses, the proliferation rates of cancerous cells and apoptosis in human cell lines [353].

Several glucosinolates and their metabolites have also been shown to have anti-cancer activity against a wide range of cancerous cells and exert their effect by several mechanisms (Table 5). High-glucosinolate-containing *B. rapa* subsp. *trilocularis* (yellow sarson) and *B. rapa* subsp. *chinensis* (pak choi) along with their low-glucosinolate-containing counterparts were tested against human colorectal cancer (CRC) cells. The high-glucosinolate-containing lines showed anti-cancer activities in terms of anti-proliferative pro-apoptotic activities, the inhibition of the nuclear factor NF-κB pro-inflammatory signaling, and ERK pathways that suppress the proliferation of cancer cells [354,355]. Many components of mustard seed were also investigated and found to have antiproliferative activity, acting to prevent or retard the spread of cells, especially malignant cells, into surrounding tissues [356,357,358] through proapoptotic effect. These components were also shown to promote or cause the apoptosis of cancer cells [358,359] and induce cell cycle arrest of the rapidly proliferating cancer cells [357,358]. Allyl isothiocyanate has the ability to prevent urinary tract cancer and was also shown to have antiproliferative activity against human prostate cancer cells [164]. Allyl isothiocyanate is also reported to exert antiproliferative activity resulting in cell cycle arrest to bladder cancer cells and apoptosis [357] as well as in vitro human colon cancer cells [360]. Phenethyl isothiocyanate (gluconasturtiin), alone [204] or combined with sulforaphane, and their N-acetylcysteine conjugates can inhibit the growth of various tumors and cancerous human and animal cell lines [204,361,362]. Benzyl isothiocyanate (glucotropaeolin) has the ability to prevent the growth of various types of cancer cells by inducing apoptosis leading to cell death at low concentrations and causing cellular necrosis at high concentrations [363,364,365,366,367,368]. Similar effects were also observed for brassinosteroid treatment on human cancer cell lines [369,370], which additionally affected cancer cell cycle progression [371]. Iberin, a glucosinolate decomposition product, has also been shown to have apoptotic potential against tumor cells, reducing the proliferation of human glioblastoma and neuroblastoma cells [175,372]. Sulforaphane, found in mustard oil [373] in combination with resveratrol, which is also a component of mustard seed [374,375], or its analog also acts as anti-tumor agent [174,376]. Enzymatic hydrolysis of glucobrassicin by myrosinase yields indole-3-carbinole (or 3-indolylmethylisothiocyanate) which further combines with ascorbic acid to form ascorbigen with stronger in vitro antioxidant activity and significantly higher in vivo activity than ascorbic acid alone, and Trolox, the two known strong antioxidant molecules commonly used as standard antioxidants [377,378]. Ascorbigen was found to prevent tert-buthyl hydroperoxide-induced cytotoxicity in human keratinocytes cultured cells to a greater extent than ascorbic acid [378].

The mustard seed proteins and protein-derived peptides were reported to exhibit in vitro and in vivo anti-cancer activities. The seed defense protein thionin [236] also has anti-cancer properties [379]. The application of thionin to cancer cells prevents the uptake of sugar, nutrients, and proteins and causes nucleotides to leak out from the cell, ultimately destroying the cancer cells [380]. Rapeseed peptides were also found to have significant in vivo free radical-scavenging ability and cytotoxic activity against human cancerous HepG2 cell lines [381], as well as liver injury, inflammation, fibrosis, and metabolic deterioration by suppressing the fibrosis-associated gene expression in mouse models [331]. Furthermore, rapeseed protein-derived anti-oxidative peptides demonstrated potent cytoprotection anti-proliferative activity against chemical-induced human cell lines and cancer cell lines through the inhibition of cell apoptosis [382,383,384,385,386].

Although it has been proposed that some isothiocyanates inhibit protein synthesis and affect carbohydrate metabolism, the activity of these compounds seems to be mainly due to the chemical reactivity of the isothiocyanate group, which binds easily to protein. Moreover, some hydrolysis products could be considered as antimitotic substances, given their strong anti-proliferative activity [155].

A lowered risk of lung, stomach, breast, prostate, pancreas, colon, and rectal cancers has been linked to the consumption of vegetables of the Cruciferae family and it is thought this may be due to the presence of active anti-cancer metabolites including glucosinolates and their breakdown products [31]. In Asia, people consume high amounts of cruciferous vegetables, up to 24% of the total vegetable intake [387], including mustard leaves and seeds [117,171,344], and they are shown to have a lower occurrence of cancer, increased life expectancy, and generally greater wellbeing than people from countries that eat less of these vegetables [10,117,171]. The regular consumption of *B. nigra* seeds improves the body’s biological defense mechanism against cancer development, and epidemiologic studies have shown that consumption reduces the rate of colon, bladder, and lung cancer [362].

### 3.5. Broncho- and Vaso-Dilating Properties

Mustard oil can be applied to the chest and back as a poultice to reduce chest congestion [388], serving to loosen phlegm, and aiding in its expulsion. Children with colds, coughs, and chest congestion are rubbed with mustard oil to alleviate their discomfort [389]. A syrup made from boiled willow bark, mullein, and rue combined with mustard oil has been described as an excellent syrup for treating cough and also eases aches and fever [390]. *S. officinale* dry plant extract has long been traditionally used to alleviate laryngitis, bronchial catarrh, and throat congestion [40].

Dry plant extracts of *S. officinale* applied to guinea pig tracheal smooth muscle were found to deactivate the histaminic, muscarinic, and cysteinyl leukotriene LTC4 receptors responsible for the contraction and triggering of inflammatory and allergic responses, and relax the smooth muscle [39]. Sinigrin (allylisothiocyanate) and sinapine in mustard may counteract the bronchial contraction [43]. These ingredients help to reduce congestion in the respiratory tract by dilating local blood capillaries, and increasing the circulation of blood to the area [391]. A mustard footbath has also been shown to move congestion out of the chest by increasing circulation to the lower part of the body, soothing deep muscle aches, and treating headaches. Mustard inhalation improves circulation to the head via the nasal passages and is used to treat chest and sinus colds [392]. However, when isothiocyanate is inhaled with oil or aspirated from mustard, it can cause airway irritation in the nasal tract in some individuals [232,393].

### 3.6. Action on Cardiovascular System

According to the World Health Organization, cardiovascular disease (CVD) is the number one cause of death globally resulting in the deaths of 17.5 million people worldwide annually [394]. Hypertension, caused by the constriction of blood vessels and high blood pressure, is one of the leading causes of CVD. Food and drugs which inhibit the angiotension 1 converting enzyme (ACE) are used to treat hypertension [395]. Leaves and seeds of the mustard *B. juncea* have been shown to have significant ACE inhibitory activity [148], and antihypertensive peptides from *B. napus* and *B. rapa* var. *campestris* have also been shown to inhibit ACE in vitro and in vivo [396,397,398,399,400]. The mechanism of inhibition was revealed from in vivo experiments whereby rapeseed protein, specifically rapakinin, Arg–Ile–Tyr, derived from the mustard 2S albumin type protein napin, caused the spontaneous dilation of the mesenteric artery in hypertensive rats via the prostaglandin-mediated prostanoid receptors. This was followed by cholecystokinin A receptor-dependent vasorelaxation [401,402]. When the blood vessels are relaxed, it causes a reduction in blood pressure, termed antihypertensive activity [402]. In addition, the oral administration of bioactive peptides, including a novel peptide Gly-His-Ser derived from rapeseed protein digest, was found to inhibit ACE and renin activities in hypertensive rats [347,398,403,404,405,406,407]. Peptides from Broccoli (*Brassica oleracea* L. var. *italica* Planch) stem and leaf protein hydrolysates were found to inhibit ACE activity [408].

Peptide concentrates derived from *B. napus* seed proteins were found to be able to prevent the development of hypertension and prevent CVD [395]. Canola protein isolate from *B. napus* showed the most prolonged antihypertensive effect in vivo, and low molecular weight peptides were found to decrease blood pressure at a faster rate [396].

Several cardiac diseases like atherosclerotic vascular disease are directly linked to oxidative stress and high blood fat levels. Superoxide anions generate pro-aggregatory isoprostanes, which contribute to an increased risk of heart attack. Mustard flavonoids can scavenge superoxide anions and lower oxidative stress. In addition, they reduce the formation of isoprostanes, facilitate anti-aggregatory PGI2 formation, and thereby exhibit anti-thrombotic action [409]. Numerous studies have suggested that flavonoids have cardioprotective and vasodilating effects, as well as acting as protective agents against coronary disease and metabolic disorders [410,411]. As edible oils, mustard oils, specifically Canola oil, contain significant quantities of unsaturated fatty acids, which can reduce cardiovascular disease. The addition of alpha-linolenic acid and other omega-3 polyunsaturated fatty acids which are abundant in rapeseed oil to the diet was proven to lower the risk of stroke and other CVDs [412,413,414]. Likewise, the oil has a notably small amount of palmitic acid compared to most other vegetable oils. Mustard and Canola oils have considerably lower profiles of palmitic acid than other plant-based edible oils (Table 2). Palmitic acid is considered as a “potential cardiovascular risk factor”, and the low profile also makes mustard and Canola oils healthier than other vegetable oils [91]. Another component of mustard oil, erucic acid was found to lower chemically induced necrosis in mice hearts [344].

*S. officinale* leaves are used to treat chest pain, cardiac weakness, and other cardiac problems and as a cardiotonic in folklore medicine [415,416,417]. The leaves were found to contain cardioactive steroidal glycosides-corchorosid A and helveticosid [39,43]. Their action is similar to the well-known cardiac glycoside digitalis which exerts its action on the heart by increasing myocardial contractability [162,415,416], relieving weak heart muscle-related complications. This may provide a scientific basis for the use of this plant in folklore medicine as a cardioactive drug.

### 3.7. Counter Irritation Action

Historically, mustard seed paste, powdered seeds, and oils have been used topically to create a counter-irritant effect by stimulating the removal of stored blood from capillaries. A counter-irritant causes cutaneous vasodilatation by stimulating the sensory nerve endings and inducing a neurogenic inflammatory response by the excitement of the capillaries of the skin or border membrane. This elicits the dilation of the blood vessels and increases the flow of blood [418], serving to block the intense pain at the site of application. The use of heat to the area of pain before applying the counter irritants effectively maximizes their pain-relieving action [419]. Glucosinolates produce a warming sensation and reddening related to the dilation of capillaries on the skin and thereby provide relief from pain [420]. Because of these properties, *B. nigra* seeds are used in the crude drug industry in a poultice inside a protective dressing [421] to apply over an inflamed, painful area of skin to improve circulation and relieve pain [422,423]. Mustard plasters are a blend of oily or waxy mixtures together with mustard seed powder alone or with other herbs and are usually placed on the chest or abdomen wherever tenderness exists. In traditional medicine, the blends were dispersed onto a cloth and wrapped tightly for storage and unrolled and applied when required [424].

To treat peritonitis, an inflammation of the serous membrane lining the abdominal cavity, a large mustard plaster was traditionally applied to the abdomen [425,426]. Likewise, to treat cholera morbus, an acute gastroenteritis with symptoms of spasmodic diarrhea, pain, and vomiting, mustard plasters were placed on the abdomen [427]. The application of mustard poultices and paste produces the quickest relief for inflammatory and related arthritic conditions by drawing out pustular constituents and toxins through the skin and dispersing the congesting blood from inside the body. The rubefacient action causes mild irritation to the skin, stimulating circulation to that area and relieving muscular and skeletal pain [428]. For endocarditis, an inflammation of the internal lining of the heart, mustard poultices to the chest were prescribed to hasten the absorption of the deposit of interstitial fluids [427].

### 3.8. Antiinflammatory and Analgesic Action

Mustards have a long history of being used in pain management [429] as poultice, cataplasms, or warm oil massage because of their counter irritant properties. Mustard seeds are used by traditional practitioners as cataplasm which is a medical dressing containing soft heated seeds that is spread on a sterile gauze or cloth and applied to the skin to treat inflamed areas to improve circulation [430]. A warm mustard poultice or oil massage can also be applied to reduce pain and inflammation providing relief from arthritis, and back, joint, muscle, and leg pain by stimulating blood flow. The heat activates the volatile allyl isothiocyanate metabolite which acts by binding directly to the transient receptor potential (TRP) ion channel receptor proteins located on the surface of animal cells responsible for detecting the sensation of high temperature, pain, inflammation and additional stimuli and sends signals to the brain [431,432]. The mechanism of action of isocyanates is similar to the mechanism of capsaicin, the active component of chili peppers. Both target the same populations of neurons, acting on TRPV1 and TRPA1 ion channels in nociceptive neurons where they trigger a signaling cascade which leads to peripheral dilation and permeabilization of blood vessels [433] and reversibly abolishes the induction of pain and inflammation in nerve cells [431].

Other compounds in the mustard extracts have also been shown to have anti-inflammatory effects [14]. Aerial parts of *B. rapa* subspecies *chinensis* (L.) Hanelt were applied to mice following induced pain using either an acetic acid-induced writhing test or tail immersion assay and the plant material was found to have significant analgesic effects [434]. Moreover, dry mustard seed powder contains a significant amount of salicylate (mono-hydroxy-benzoates, around 26 mg per 100 g powder) [435], a metabolite of aspirin widely used for pain, fever, and inflammation [435,436]. The high content of selenium and magnesium in mustard oil could also account for anti-inflammatory activity. The oil stimulates sweat glands assisting to decrease temperature when the body is inflamed [242]. The aerial parts of *Erysimum corinthium* were reported to possess triterpenes-amyrin which exerts antinociceptive and anti-inflammatory properties by the activation of cannabinoid receptors and inhibition of the production of pain mediators like cytokines and expression of NF-κB, CREB, and cyclooxygenase 2 in mice [437,438] as well as lupeol which has strong anti-inflammatory, antiarthritic properties [437,439]. Flavonoids can inhibit cyclooxygenase and lipoxygenase enzymes and thus reduce the formation of pro-inflammatory pain mediators (prostaglandins, leukotrienes, reactive oxygen species, and nitric oxide) and strongly inhibit prostacyclin production [409]. *Sinapis alba* seed extract was found to reduce inflammatory infiltrations of T-cells, dendritic cells, macrophages, nuclear factor (NF)-kB, interferon, and interleukin in experimental mice cells; reduce some inflammatory mediators at the level of mRNA and protein [440]. As mentioned previously *B. nigra* seeds were also found to raise the high-density lipoprotein cholesterol (HDLC) level [316] which has anti-inflammatory properties [441]. A regulatory protein NPR1 in *B. juncea* can inhibit NF-κB, a family of dimeric transcription factor protein complexes which regulate inflammation and are linked to cancer, inflammatory, and autoimmune diseases [353]. Another mustard metabolite, sulforaphane was also found to inhibit the NFκB complexes [440]. Sulforaphane is also reported to induce anti-inflammatory effect and increase the local antinociceptive actions of morphine [442]. Indole-3-carbinol, a mustard component, can act as a ligand for the aryl hydrocarbon receptor which has the ability to influence both immune-regulatory T cell and T-helper 17 cell differentiation, inhibit delayed-type hypersensitivity by inducing a shift from pro-inflammatory T-helper 17 cells to regulatory anti-inflammatory T cells and thus play an important role against inflammation [197]. Cruciferins, one of the major seed storage proteins [73], from rapeseed (*B. napus*) and radish (*Raphanus sativus*) showed significant in silico overall occurrence frequency for DPP-III inhibitory peptides which suggested that they may have high potential to possess an antinociceptive activity [443].

### 3.9. Antiarthritic Action

The use of mustard to treat arthritis is approved by Commission E in Herbal Medicines, a German-based European scientific advisory body which approves substances and products used in traditional, folk, and herbal medicine [43]. Osteoarthritis is associated with joint pain and characterized in part by defective articular cartilage [440,444]. The lack of curative drugs to treat osteoarthritis means that the management of the disease, by the use of bioactive phytochemicals, offers a novel and attractive approach to preventing the onset and/or progression of the disease [440]. Sulforaphane in mustard seeds was found to impact articular cartilage in experimental osteoarthritis models, reducing the cytokine-induced expression of proteinase enzymes that degrade cartilage from chondrocyte cells; in addition, it inhibited the cytokine-induced degradation of cartilage explants; and ultimately, it diminished cartilage destruction caused by osteoarthritis [373,440].

### 3.10. Diuretic Action

Mustards have been traditionally used as diuretics [21]. In general, diuretics help turn excess body water into urine, interfering with normal kidney function by reducing the amount of sodium and water absorbed back into the bloodstream [445]. Glucosinolates of mustard seeds cause irritation to the mucus tissues of the renal system producing a diuretic response. Mustard seeds contain crude mucilage which accounts for 5% to 20% of the seed depending on the species [446]. Usually, this mucilage contains 80% to 94% carbohydrates [13]. These carbohydrates also play a role in diuresis. For instance, *B. juncea mucilage* was found to raise serum glucose levels, preventing the rise of creatinine levels in urine and ultimately increasing the volume of urine in animal studies [317]. Mustard, rapeseed, and Canola seeds also contain mannitol (0.7 mg/g dry weight) [447] which acts as an osmotic diuretic. When mustard is ingested in large amounts, the mannitol is absorbed slowly from the gastrointestinal tract into the blood stream, increasing the volume of fluid [436]. The presence of mannitol causes the retention of water to balance the osmotic pressure of the urine. The saponins and vanillin present in the seed also have diuretic properties [448].

### 3.11. Anthelmintic Activity

Different mustard seeds and oilcakes were found to diminish the population of plant parasitic nematodes in the soil when cultivated and are used in agriculture as rotation crops to eliminate these pests [449]. Mustards have also been found to have anthelmintic activity in humans and animals, helping to expel internal parasitic worms. Seeds and purified secondary metabolites—sinigrin, sinalbin, gluconapin, epi-progoitrin, glucoiberin, and glucoerucin, were tested on a simulated human gastrointestinal tract infected with an earthworm (*Pheretima posthuma*) model as well as other nematodal models and found to have dose-dependent anthelmintic potential [177,450,451]. The seeds were found to paralyze (vermifuge action) and eventually kill the earthworms (vermicidal activity). However, compared to a standard anthelmintic drug, mebendazole, the mustard seeds required a prolonged time to kill the worms at the dose of 20 and 40 mg/mL [452]. Additionally, volatile compounds released from mustards prepared as infusions have been shown to be effective for the expulsion of tapeworm but ineffective against hookworm, roundworm and whipworm [450].

Mustard secondary metabolites may exert their anthelmintic activity by several mechanisms. They can produce end-products such as phenolics, ammonia, fatty acids, and hydrogen sulfide which are directly toxic to the nematodes [453,454]. In previous studies, it was also revealed that tannins [367] and the polyphenolic compound ellagic acid [368] are present in mustard seeds in adequate amounts to produce anthelmintic activity in a large dose. Additionally, mustard seed and oilcake preparations increase the activity of bacteria and fungi which are antagonistic to parasitic nematodes [455,456].

Mustard seed is one of the ingredients of the commercial product Kochi free (Amber technologies) used to control avian coccidiosis, a disease of birds and mammals that chiefly affects the intestines, caused by coccidia, a single-celled parasite [457].

### 3.12. Emetic Action

Emetics are used to expel toxins from food or other poisonings by inducing vomiting by either acting on the vomiting nerve center of the brain or locally by irritating the nerves of the gastric mucus membrane. In the case of mustard, several isothiocyanate products (methyl isothiocyanate, allyl isothiocyanate, and phenyl isothiocyanate) act as irritants on the nerves of the stomach and mucus membrane [249] which consequently results in immediate nausea, vomiting, and colic [450]. When mustard seeds are eaten raw, trypsin inhibitor proteins can also irritate the human digestive system [292,458] which may also lead to vomiting.

### 3.13. Laxative Action

Sufficient ingestion of mustards stimulates gastric mucosa, increases pancreatic secretions, and causes irritation to the gastrointestinal tract which decreases the amount of time food remains in the intestine and consequently reduces water absorption. This results in a higher-than-normal release of water in the feces causing laxation [265]. Additionally, daily intake of mustard seeds as a condiment increases the blood circulation to the bowels, increasing the rate of transit of food through the gastrointestinal tract and decreasing the absorption of liquid from the food residue [392]. *B. juncea* seeds have been reported to alter hepatic functions increasing the secretion of digestive enzymes into the gastrointestinal tract and increasing digestion. Moreover, mustard seeds have an abundance of mucilage (2–5% of dry seed weight) [13,459], composed of an extremely hydrophobic pectin-rich polymer, which serves to increase viscosity and volume of the stool and acts as a laxative against constipation [266]. *Sinapis alba* seeds contain hydrogen sulfide which irritates the skin and mucous membranes [460]. The seeds also have a minor amount of trypsin inhibitor proteins [292], which prevents the gastric enzyme trypsin from digesting proteins in the small intestine [458]. When seeds are eaten raw, trypsin inhibitor proteins present can irritate the human digestive system which can also cause laxation [292,458].

### 3.14. Development of Body Muscle and Aphrodisiac Action

Traditionally, mustards were used to develop body mass and as an aphrodisiac [268]. Brassinosteroids and isocyanates stimulate local capillaries and improve the blood flow at the site of application. For this reason, eating mustard seed and application of mustard oil are considered as an aphrodisiac and used to treat erectile dysfunction in some traditional practices [30,279,461]. Extract of *B. rapa* administered orally was found to increase sperm motility and volume in an animal reproductive system model [268]. Mustards also produce aphrodisiac action by altering serum testosterone levels, and smooth muscle relaxation in the male genitalia arbitrated by a spinal reflex. They can also induce androgenic effects such as the development of muscle, increase blood pressure, and raise water/salt retention in tissues and blood in animals [147]. Mustard brassinosteroids are structural analogs of cholesterol-derived animal steroids, which promote growth regulation activity in plants and can have an anabolic effect in humans by increasing protein synthesis and lowering the breakdown of muscle proteins without coupling to the androgen receptor [462].

### 3.15. Treatment of Baldness

The application of mustard oil onto the skin surface increases blood flow and causes vasodilation. Therefore, it has been used as a remedy to treat baldness by applying topically on the skull [445], and on skin to revitalize body hair growth [463]. Traditionally, it is recommended by herbal practitioners to rub the oil onto the scalp twice a day to increase hair growth and prevent further hair loss. The action of mustard in alopecia (hair loss) is quite similar to that of minoxidil, an over-the-counter approved drug by the Food and Drug Administration (FDA) to treat hair loss. Both act by stimulating cell proliferation in the hair follicles on the scalp by providing increased blood flow and the stimulation of prostaglandin synthesis [464]. Dandruff related to fungal infection on the scalp is another cause of hair loss and baldness. The anti-fungal activity of mustard oil helps to prevent fungal infections [465].

### 3.16. Effects on Blood Flow and Bodily Secretions

Mustard has a secretory effect on several organs and glands such as salivary, mucus, milk, sweat, and bile glands, as well as increasing blood flow to the site of application [256]. It has traditionally been used as a poultice of bread, milk, and mustard applied on the breasts to promote lactation when a lactating mother’s milk is low [256].

Mustard seeds have also been found to increase the secretion of saliva in the mouth and bile acids in the intestine and thus act as a digestive stimulant. Because of these properties and as mentioned above, it has traditionally been used as an appetizer [276,277,466]. The ingestion of garlic mustard leaves causes or increases sweating by opening the sweat ducts [122].

Among the secondary metabolites of mustard, only the allyl isothiocyanate has been reported to have a negative effect on secretion. This metabolite was found to directly inhibit the (H^+^ + K^+^)-ATPase and thus reduce the acid secretion in the membrane of the parietal cells in the stomach [467].

In Asia, it is a common practice to massage a newborn with mustard oil and the oil massage has been found to improve blood flow, blood vessel diameter, and ultimately induce sleep, and increase the growth of the newborn [388,468].

### 3.17. Reduction in Fat and Body Weight

Traditionally, mustard is used to burn body fat and reduce body weight [469]. Rapeseed proteins and their enzymatic hydrolysis products were found to inhibit fat accumulation in the murine mesenchymal stem cells line in vitro, similar to or slightly less than that of the widely used anti-obesity drug Orlistat which was used as a control [470]. In another study, an anti-hypertensive peptide derived from rapeseed napin protein Arg-Ile-Tyr, called Rapakinin, was found to have anorexigenic activity, suppressing the appetite and lowering the intake of food and gastric emptying time in a rodent model [399]. It was suggested that the Arg-Ile-Tyr protein acted by stimulating the secretion of the peptide hormone cholecystokinin [471]. To elucidate the underlying molecular mechanism of anti-obesity activity, *Brassica juncea* extracts were applied to in vivo high-fat diet-induced obese mice [472,473,474] as well as in vitro pre-adipocyte cells [472]. Results suggested that the oral administration of the extracts lowered the body weight, improved liver damage, and inhibited lipid accumulation in high-fat diet-fed obese mice [472,473] compared to the standard anti-obesity extract of *Garcinia cambogia* extracts with anti-obesity effects [475]. In particular, the extract reduced the mesenteric, epididymal, and total adipose tissue weights as well as the bio-markers of obesity in blood namely serum triglyceride level, total cholesterol profile, and high-risk-factor low-density lipoprotein (LDL) cholesterol and increased the healthy high-density lipoprotein (HDL) cholesterol [474]. The in vitro assay on preadipocyte cells indicated that the extract reduced the expression of the development and accumulation of storage fat tissues as well as lipid synthesis proteins. Similar studies on high-fat diet-fed obese mice models indicated that feeding rapeseed diacylglycerol oil improved the serum obesity-related indices and lipid metabolism in the fat storage organs including adipose tissue, liver, and intestine [476]. In addition, the extract escalated the expression of heat generation and fatty acid oxidation proteins [473,474].

### 3.18. Antidepressant Effects

Traditionally, the application of mustard seed oil to the skin surface is believed to have a soothing and relaxing effect. Constituents of rapeseed, including eicosapentaenoic acid, and other omega-3 fatty acids regulate the hypothalamus–pituitary–adrenal axis activity in the brain and lower cytokines involved in inflammation [477] reducing stress. An extract of *B. rapa* subspecies *chinensis* (L.) was tested for antidepressant activity and was found to have significant antidepressant activity compared to a control [434]. Several antidepressants have the ability to enhance glucocorticoid receptor function [478] which controls a number of physiological functions like intermediary metabolism, skeletal growth, the performance of the immune system and cardiovascular system, reproductive system, and cognitive functions [479]. The rapeseed peptide, rapakinin, was also reported to have vasorelaxing activity [402].

### 3.19. Use in Adrenoleukodystrophy and Adrenomyeloneuropathy

Two components of mustard oil, erucuic acid and oleic acid, are constituents of Lorenzo’s oil [344,480], which is used to treat associated inherited genetic disorders of the nervous system, namely adrenoleukodystrophy found in children; and adrenomyeloneuropathy, which arises in adults, both linked to the accumulation of saturated very-long-chain fatty acids in the brain white matter, adrenal glands, fibroblasts, and plasma [344,480]. The application of Lorenzo’s oil inhibits the synthesis of very-long-chain fatty acids as well as limits the accumulation of saturated long-chain fatty acids in the brain [480,481] and in the blood [482]. Docosahexaenoic acid (DHA) in rapeseed oil is a major component of the nervous system including in the brain and eye tissues and supplementation of DHA and antioxidants are thought to be two of the reasons that improve nervous system disorders [483].

## 4. Adverse Effects and Other Antinutritional Effects of Mustard

In addition to the beneficial effects of mustards for human health, there is also evidence that they can cause some adverse effects, ranging from minor allergenic reactions to life-threatening anaphylactic shock to cardiac arrest. However, these reactions vary greatly depending on the dose and from individual to individual.

### 4.1. Cardiac Effects

*S. officinale* seeds contain the steroid glucosides corchorosid A and helveticosid, which are utilized to treat cardiac dysfunction by increasing myocardial contractability [39]. However, the consumption of large amounts of leaves and seeds of *S. officinale* (greater than 5 g per day) can affect the cardiovascular system by hyperactivating the heart leading to palpitations and even causing death depending upon the dose and the medical history of the patient [43,460].

### 4.2. Allergic Effects

Mustard seeds are recognized by the human immune system, triggering an IgE-mediated allergic reaction [73,484,485]. In an allergic reaction, the immune system produces antibodies to fight seemingly harmless substances, this causes a range of responses from minor irritation, and gastrointestinal upset, to the more serious anaphylaxis [73]. Mustard/rapeseed intolerance or IgE-mediated allergy is the most common among the allergies associated with foods, accounting for about 7% of allergic patients globally [486,487]. Mustard seeds contain considerably high amounts of indigestible 2S albumin-type napin proteins which are not digested by pepsin in the human digestive tract and remain intact [488]. The undigested protein can induce an immediate immune response in the gastrointestinal tract [489] by increasing the levels of specific IgE antibodies and initiating the response of inflammatory mediators producing antigen–antibody reactions [487], and in severe cases can result in anaphylactic shock leading to death.

Among European countries, France is the largest producer and consumer of mustard products, and after eggs, peanuts, and cow’s milk, mustard is the fourth most significant allergen [90]. Conversely, mustard has been reported to reduce the allergenic response caused by contact dermatitis by lowering the quantity of infiltrating Langerhans cells in tissue and suppressing the expression of lymphocyte activating factor, tumor necrosis factor alpha, and interleukin (IL)-6 mRNA [14,352].

### 4.3. Goitrogenic Effects

Mustard seeds and rapeseeds contain glucosinolates bound to glucose (Section 2.2, Table 2) which break down into the goiter-causing agent goitrin by the enzymatic action of myrosinase (Figure 2). The consumption of Brassicaceae plant products may result in the lowering of thyroid function and enlargement of the thyroid gland due to the presence of these goitrogenic glucosinolates and their metabolites epigoitrin, progoitrin, napoleiferin, and thiocyanate [464].

In humans and animals, the dietary micronutrient iodine facilitates the production of thyroid hormone, the hormone necessary for normal metabolic function and growth. Some glucosinolate metabolites namely goitrin and gluconapoleiferin inhibit the thyroperoxidase enzyme which leads to the decreased production of the thyroid hormone, resulting in the enlargement of the thyroid tissue which is the disease goiter [490]. Another glucosinolate hydrolysis metabolite, thiocyanate, decreases iodine uptake by the thyroid. As a consequence, the synthesis of the thyroid hormone is reduced [491]. To minimize this, the thyroid gland has to increase its function and become enlarged [492]. Selective breeding of Brassicaceae mustards has lowered the amounts of glucosinolates, for example, conventional rapeseed meals can have 3–8% glucosinolate whereas Canola contains less than 0.2% [493] which is a level below that causing physiological effects associated with thyroid problems.

### 4.4. Hematological Effects

*B. oleracea* has been reported to have hematological effects—antiplatelet and hypocholesterolemic activities in experimental rabbits [494]. High ingestion of Brassica vegetables including rape, kale, and raphnobrassica may cause the accumulation of high concentrations of nitriles in cattle blood which might reduce the ability of carrying oxygen capacity and thus there is a shortage of sufficient healthy red blood cells or hemoglobin in blood to carry oxygen to the tissue. This condition is termed as ‘kale anemia’, ‘red water’, or specifically ‘hemolytic anemia’ in cattle [477]. Small to moderate ingestion can cause scours, salivation, abdominal pain, abortions, staggers, and convulsions while a large amount of ingestion can often result in sudden death in gazing cattle [495,496]. *B. rapa* var. *campestris* can accumulate toxic quantities of s-methyl-L-cysteine and s-methyl-L-cysteine sulfoxide (SMCO) in the aerial organs including flowers, leaves, and stems. After ingestion, rumen microorganisms transform this to toxic dimethyl disulfide which causes hemolytic anemia in livestock. Consumed in large amounts, SMCO (more than 1% of dry matter of animal feed) can be toxic to certain livestock [497,498]. However, there is a dearth of reports on the hematological effects of brassica vegetables on human health probably because they do not eat those in bulk amounts and eat mostly after cooked.

### 4.5. Neurological Effects

Thiamine (vitamin B1) is a coenzyme necessary for the metabolism of carbohydrates, fats, and proteins [499]. The intricate chemical composition of Brassica vegetables causes thiamine depletion in ruminants. In addition, Brassica vegetables are rich in sulfur. The alteration of thiamine level and concurrent exposure to sulfur cause the necrosis of gray matter of the cerebral cortex in the brain and swelling in the ruminants known as polioencephalomalacia [499]. In this neurological disease, cattle may experience sudden onset of blindness, muscle tremors and convulsions, head-pressing, star gazing), rapid eye movements, incoordination, recumbency, and later, death. If the clinical signs could addressed at the early stage, the toxicity could be successfully treated with vitamin B1 [496,499].

## 5. Possible Application in Pharmaceutical Manufacturing

Rapeseed oil and its metabolites have the potential to be used as active ingredients and excipients for pharmaceutical products. As an example, rapeseed oil was found to enhance in vitro platelet aggregation and thromboxane production [85]. Thromboxane is one of the main cyclooxygenase products of human platelets and a potent vasoconstrictor which enhances the activity of platelets [500]. Patients who have bleeding disorders or suffer from thrombosis are treated with substances that boost in vitro platelet aggregation and thromboxane production and for this, rapeseed oil could have possible medicinal applications. The absorption and binding pattern, sensitivity, and efficacy of the mustard oil could be tested on the skin surface and also with human serum [501] for the identification of sensitivity patterns and could be used topically to stop bleeding at the site of hemorrhaging [409,502].

In Asian traditional herbal medicine, rapeseed oil is blended with active painkilling ingredients to produce a balm to treat acute traumatic pain [503]. The mustard seed essential oil could be a safe and favorable vehicle also for the formulation of anti-microbial compounds in soft gelatine microcapsules because it has strong bactericidal properties and mustard oil is found to provide strong chemical stability under a range of relative humidity and temperature when encapsulated in complex coacervation microcapsules. The presence of mustard could also help to harden the microcapsules depending on the selection of the coacervate microsphere [157]. The use of protein and other components as active ingredients and excipients is still marginally developed [4]. Drugs which are required to stay longer in the stomach for maximum effect can be formulated with the addition of mustard seed fibers which would allow for prolonged gastric exposure (as mentioned in the previous section). Research found that the major rapeseed protein cruciferin is an ideal material as a carrier molecule for the delivery of poorly soluble, hydrophobic, and less bioavailable bioactive compounds like curcumin. Therefore, cruciferin could be used as a drug vehicle in nanoencapsulation because of its amino acid composition resistance to gastric degradation; biocompatibility and good emulsifying and gelling properties; formation of homogeneous, fairly dense spherical and stable structures in colloid systems; and non-toxic nature to other drug carriers and vehicles as well as enhancing the cellular transport of active ingredients [504,505,506,507,508]. It was effectively used in curcumin delivery systems as a polyelectrolyte complex nanocarrier in encapsulated curcumin preparations and it was found to increase the bioavailability of curcumin [507].

## 6. Summary and Outlook

The potential health benefits of mustards are enormous. Although the chemical characterization of metabolites present in mustards has been performed, more studies are needed on the individual components and their bioactivity. Where analyses have been carried out, experimental evidence is mostly based on in vivo rat and mice models; however, rodent models are limited in their ability to mimic the extremely complex processes of human physiology. For example, differences in the action of sulforaphane and neoglucobrassicin on the induction of the quinone reductase enzyme and the inhibition of nitrite production were detected when these metabolites were tested in human and rodent cell lines. Neoglucobrassicin was found to inhibit the effect of sulforaphane on the quinone reductase enzyme in human hepatoma cells but not in rodent hepatoma cell lines [509].

Another example showing the importance of using correct models is related to erucic acid toxicity. In the United States and many European countries, consumption and import of mustard oil are restricted due to the high erucic acid profile. While no adverse effects have been reported for human exposure to erucic acid, acceptance levels have been determined based solely on animal models [54], with no direct analysis of acceptance levels for human exposure. To date, there are no known human clinical trials conducted to evaluate the therapeutic activity of the individual chemical components of mustard.

A mixture of multiple plant secondary metabolites can have synergistic effects but also can even antagonize the activity of each other [321]. For instance, the effect of iberin on modifying phase I and phase II detoxification enzymes in experimental rats was shown to have an increased effect when combined with a mixture of glucosinolate derivatives than when given alone [152]. Complexation of some known drug molecules with the natural mustard secondary metabolite can also enhance or synergize the activity of the secondary metabolite or the activity of both of the molecules. For example, the complexation of allyl isothiocyanate with the known painkiller drug celecoxib, which alone has no evident anti-cancer activity, was found to exert greater inhibition of urinary bladder cancer growth and muscle invasion in a rat bladder cancer cell line than that of either allyl isothiocyanate or celecoxib alone [156]. To explore the effects of the chemical modification of individual natural mustard secondary metabolites, effects of concurrent administration or complexation with other molecules or chemical groups, more studies, including structure–activity relationship (SAR) and quantitative structure–activity relationship (qSAR) analysis and in silico approaches need to be carried out. Lots of work is still needed before the phytoconstituents of mustard seeds can make their way from traditional folklore uses into the modern mainstream medical domain.

## 7. Conclusions

The diverse components of mustards have been employed against a wide range of altered health conditions since their domestication in ancient times up until the contemporary days. However, the current scientific knowledge regarding the phytochemical composition and biological activities associated with Brassicaceae oilseeds and mustards, insights into their mechanism of action, and potential therapeutic applications is limited. The medicinal properties of these mustard species can be attributed to the diverse array of bioactive compounds they produce, including sulfur-containing glucosinolates, omega-3 fatty acids, terpenoids, phenylpropanoids, flavonoids, tannins, S-methyl cysteine sulfoxide, and trace elements. This study underlines and links traditional knowledge with contemporary scientific approaches to bridge the gap between ancient medicinal practices and modern healthcare solutions. The diversity in chemical nature and variety of bioactivities make them intriguing starting templates for the development of novel therapeutics. The information on the bioavailability and pharmacokinetics could be investigated using the latest technologies like UPLC-MS/MS, GC-MS/MS, and NMR-based metabolomics, and advanced animal and cell models could be used to determine the safe dose for human health benefits and ensure rational dosage regimen.

## Figures and Tables

**Figure 2 ijms-25-09039-f002:**
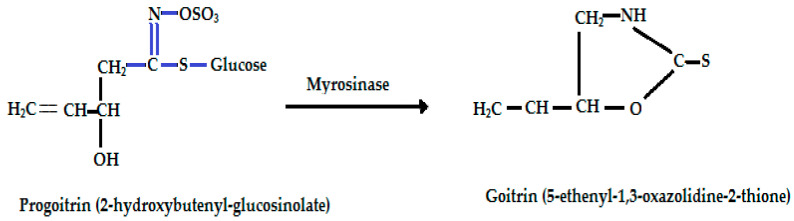
Enzymatic hydrolysis of progoitrin by the action of myrosinase yields the potent goitrogenic metabolite goitrin. The same reaction happens in the intestine by the action of bacterial thioglucosidases.

**Table 1 ijms-25-09039-t001:** Different mustards in Brassicaceae species.

Scientific Name	Common Name	Habitat	References
*Alliaria petiolata* (M. Bieb.) Cavara and Grande, Syn: *Alliaire officinalis*	Garlic mustard, hedge garlic mustard, and jack-by-the-hedge mustard.	A biennial herb, native to various regions of Africa, temperate and tropical Asia, and Europe, and naturalized to the United States and Canada.	[5,6,7]
*Brassica rapa* var. *campestris* L.	Field mustard, wild turnip mustard, bird rape, turnip rape, and winter mustard.	A winter annual or biennial plant native to central Asia and Europe.	[8,9,10,11]
*Brassica carinata* A. Braun.	Ethiopian mustard and Abyssinian mustard.	Traditional African vegetable cultivated in Ethiopian highlands.	[7,12]
*Sinapis alba* Boiss. (Hook f. & Th.), Syn: *B. hirta* Moench, *B. alba* Linn.	White mustard, yellow mustard, and rai mustard.	Best known mustard in Europe, grown in the Mediterranean region, India, and China.	[13,14]
*Brassica juncea* (L.) Czerniak. Coss., Syn: *B. integrifolia*	Brown mustard, Asian mustard, Oriental Mustard, Chinese mustard, Indian mustard, leaf mustard, giant red, sarepta mustard, Asiatic mustard, mustard green, and wild Brazilian mustard.	An annual herb native to Eastern and Southern Asia. It is widely cultivated throughout India, central Africa, south Russia, and steppes of the northeast Caspian Sea. It is a natural hybrid of *B. rapa* and *B. nigra*.	[15,16,17,18,19]
*Brassica napus* (var. napus or var. oleifera)	Rapeseed, oilseed rape, colza, rutabaga, swede, Swedish turnip, leaf rape winter oilseed rape, and summer rape.	Native to South-East Asia and Eurasia and is most commonly grown in northern temperate regions including northern Asia, Japan, Korea, Northern China, Scandinavia, and Russia. Dominant species in Europe.	[8,20,21,22,23]
*Brassica nigra* (L.) W.D.J. Koch. Syn.: *Sinapis nigra*, *B. sinapioides*	Black mustard, brown mustard, cadlock, Scurvy mustard, Senvil mustard, short-pod mustard, true mustard, and warlock.	An annual herb native to most parts of Europe, the Mediterranean region, and other parts of north Africa, and has been naturalized in Great Britain and North America.	[15,24,25,26,27]
*Brassica rapa* L.	Oilseed rape, rape oilseed, kale rape, rapa, rappi, keblock, colza, bird rape, rape mustard, field mustard, or rapeseed.	The most cultivated oilseed in many countries like Australia, Canada, and Poland.	[28,29]
*Calepina irregularis* (Asso) Thellung.	White ball mustard, smooth ball mustard.	Native to the countries in central Asia, the Mediterranean basin and naturalized in central Europe, North America, and Australia.	[30,31]
*Camelina sativa*	Camelina, false flax, wild flax, gold-of-pleasure, linseed dodder, German sesame, and Siberian oilseed.	Central Asia and northern Europe	[32,33]
*Erysimum corinthium*	Wormseed mustard and wallflower.	Native to temperate Eurasia (Southeastern Europe), North Africa and Macaronesia, and North America south to Costa Rica.	[34]
*Erysimum repandum* L.	Spreading wallflower, bushy wallflower, and spreading treacle-mustard.	NF	
*Neslia paniculata* (L.) Desv. Syn.: *Myagrum paniculatum*	Ball mustard, common ball mustard, yellow weed, neslia, neslie, or moutarde.	Native Eurosiberian southern-temperate species commonly naturalized in South Australia, Canada, Europe, and Asia, South America.	[35,36]
*Sisymbrium officinale* L. Scop., Syn.: *Erysimum officinale*	Erysimum, English watercress, mustard, hedge mustard, St. Barbara’s Hedge Mustard, common hedge, singer’s plant, and thalictroc.	An annual or biennial mustard found on roadsides and wastelands as a weed of arable land in Eurasia, the Mediterranean area, northern Africa, Scandinavia to north Africa, and Asia and naturalized in New Zealand.	[37,38,39,40,41]
*Sisymbrium orientale* Syn.: *S. columnae* Jacq.	Asian hedge mustard, Indian hedge mustard, eastern rocket, oriental wild rocket, and London rocket.	Native to Europe, Asia, and north Africa, and further introduced in much of the rest of the world, including parts of North America and Australia.	[37,42,43]
*Sisymbrium erysimoides* Desf.	Smooth mustard, and Mediterranean rocket.	A desert plant native to Middle Eastern Arab countries and naturalized in Australia, North America, and New Zealand.	[44,45,46]

Syn.: Synonym; NF: Information not found.

**Table 2 ijms-25-09039-t002:** Total oil content, fatty acid composition, and total polyphenol content of different Brassicaceae mustards.

Fatty Acid	*Alliaria petiolata*	*Sinapis alba*	*B. carinata*	*B. juncea*	*B. napus*	*B. nigra*	*B. rapa*	*Calepina irregularis*	*Camelina sativa*	*Erysimum repandum*	*Neslia paniculata*	*Sisymbrium erysimoides*	*S. officinale*	*S. orientale*	Canola (low-erucic acid) *B. napus*
Total oil content (% od seed weight)	15–30	25–36	30–47	25–45	40–50	21–40	25–45	16–36	38–43	20.1–40	15–25	30	7–12	22–30	33–50
	Saturated fatty acids														
Palmitic acid (C16:0)	3–4	2–7	2–12	2–4.5	2–7	2–4	2–4.5	0–15	4–8	7.0	3–17	10–15	2–8.0	8–10	3.3–6
Stearic acid (C18:0)	0.3–0.5	0.1–3	2–3.5	1–2	0.5–2.5	1–2	1.5–2.5	2–3	2.1–2.3	2.0	1–4	0.5–1	0.3–1.5	0.1–1	1.1–2.5
Behenic acid (C22:0)	ND	0.5–1	ND	0–1	0–1	ND	ND	ND	0.3	0.7	ND	ND	ND	ND	~2.0
Lignoceric acid (C24:0)	ND	ND	ND	0–1	0–1	ND	ND	ND	ND	ND	ND	ND	ND	ND	~0.2
Arachidic acid (C20:0)	0.4–1	0.5–1	ND	0.5–1.5	0.1–2.5	0.5–1.5	ND	ND	1–1.8	2.0	ND	ND	ND	ND	0.2–0.8
	Unsaturated fatty acids														
	Monounsaturated fatty acids														
Palmitoleic acid (C16:1)	ND	0.1–0.2	0.1–0.2	0.2–0.5	0.2–1.5	0.1–0.2	0.1–2.0	ND	0.07–0.1	ND	ND	ND	ND	ND	0.1–0.6
Oleic acid (C18:1)	6–9	15–39	7–19	8–33	8–63	9–23	11–62	0.1–2	12.7–14.8	7.0	9–36	10–15	1–7.5	3–7	52–66.9
Gadoleic/eicosenoic acid (20:1)	4–5	6–12	5–9	5–12	1–13	5–8	1–11	ND	11.1–12.8	9.0	ND	3–4	ND	1–4	ND
Erucic acid (C22:1)	41–49	23–51	35–45	22–55	14–55	27–46	9–51	3–7	2.3–3.5	17.0	0.1–1	15–20	8–17	11–18	0.3–1.6
Nervonic acid (C24:1)	6–8	1.2–3	ND	0–2.5	0–2	ND	ND	0.1–1	ND	1.0	0.1–0.3	ND	0.5–1.5	1–0.6	ND
	Polyunsaturated fatty acids														
Linoleic acid (C18:2)	18–22	7–33	15–22	12–21	12–21	6–17	10–13	ND	2.4–24.3	17.0	ND	15–20	ND	10–14	16.1–28.8
Eicosadienoic acid (C20:2)	0.5–1	ND	ND	0.3–1	0.5	3–4	ND	ND	11.1–12.8	2.0	ND	ND	ND	ND	~0.1
Linolenic acid (C18:3)	4–7	8–20	16–20	8–20	8–14	3–15	1–17	7–26	33.7–36.9	33.0	4–10	20–31	18–30	35–42	6.4–14.1
Roughanic acid (C16:3)	ND	ND	ND	ND	0.7–1	ND	ND	ND	ND	ND	ND	ND	ND	ND	ND
Other fatty acids	1.7	0.1–2.2	ND	6–8	ND	ND	ND	ND	ND	ND	ND	ND	ND	ND	ND
Total MUFA	ND	72.19	62.49	66.56	ND	60.8	ND	46.5	30–36	ND	ND	38.4	39.8	ND	ND
Total PUFA	ND	2.05	32.65	29.59	ND	31.5	ND	39.6	50–60	ND	ND	38.9	47.2	ND	ND
Total SFA	ND	4.48	4.57	3.82	ND	6.7	ND	13.5	13.3	ND	ND	21.4	13	ND	<7
TUFA/TSFA	ND	ND	ND	10.4	ND	13.8	ND	6.5	ND	ND	ND	3.6	6.7	ND	ND
Total polyphenol content	ND	72.56	23.5	21.02	47.48	ND	ND	ND	ND	ND	ND	ND	ND	ND	ND
References	[92]	[93]	[93]	[93,94]	[95]	[94]	[86]	[31,94]	[96,97,98]	[35]	[94]	[35]	[38,39,94]	[99]	[63,100,101,102]

Values are % on dry weight, oil content is expressed of % dry weight, and total polyphenol content is expressed as mg/g gallic acid equivalent and varies with cultivars, genotype, tissues, and plant age [12,86,93,103]; ND = not detected.

**Table 3 ijms-25-09039-t003:** Vitamins that have been reported in Brassicaceae mustard leaves (per 100 g edible portion).

Vitamin	Mustard Species (Amount per 100 g)	References
Vitamin A/carotenoid	*Alliaria petiolata* (contains more vitamin A than spinach, 13.3 mg), *Sianpis alba*, *B. juncea* (27 mg), *B. nigra*, *Sisymbrium erysimoides*	[16,19,35,44,122,123,124,125]
Vitamin B Complex		[126]
Thiamine	*B. napus* (0.8 mg)	[126]
Riboflavin	*B. napus* (0.3 mg)	[126]
Niacin	*B. napus* (8.1 mg)	[126]
Pyridoxine	*B. napus* (1.9 mg)	[126]
Vitamin C/Ascorbic acid	*Alliaria petiolata* (contains more vitamin C than orange—261 mg), *B. nigra*, *Erysimum repandum*, *S. officinale*, *B. juncea* (72–89 mg)	[6,122,123,124,125,127], USDA National Nutrient data base
Vitamin E/tocopherols (α-, β-, γ-, and δ-tocopherol, γ tocopherol is predominant)	*Sinapis alba*, Canola (*B. napus*) seed oil: (α tocopherols 12 mg, γ tocopherols 21.3 mg)	[35,128]
Vitamin K	All rapeseeds including *B. juncea* (0.3 mg). Canola seed (*B. napus*) oil (70–150 µg)	[16,129], USDA National Nutrient data base, [130,131]

**Table 6 ijms-25-09039-t006:** Traditional/folkloric use of mustard extracts.

Traditional Use	Mustards	References
Anti-microbial activity	*Alliaria petiolata*, *Sisymbrium officinale*, *S. erysimoides*, *B. hirta*, and *B. nigra*	[6,39,123,158,216,250,251,252]
Antidiabetic activity	*B. juncea*, and *B. nigra*	[253,254]
Treatment for vitamin C deficiency	*A. petiolata*, *B. rapa*, *B. napus*, and *Erysimum repandum* are antiscorbutic	[6,21,123,255]
Diuretic activity	*A. petiolata*, *B. juncea*, *B. napus*, *B. nigra*, *B. rapa*, *S. officinale*, and *S. orientale*	[6,21,38,40,41,123,255,256,257,258,259]
Expectorant activity	*A. petiolata*, *S. orientale*, *S. officinale*, and *S. erysimoides*	[6,38,40,41,258,259,260,261]
Stimulant activity	*A. petiolata*, *S. alba*, *B. juncea*, and *B. nigra*	[6,256,257,262]
Analgesic activity	*B. rapa* var. *campestris*, *B. juncea*, *B. napus*, *S. erysimoides*, *S. officinale*, *B. carinata*, *Neslia paniculata*, and *Calepina irregularis*	[21,44,123,160,242,256,263,264]
Activity in cold and flu	*Sinapis alba*, *S. officinale*, *S. erysimoides*, *B. napus*, and *B. nigra*	[39,251,258,260,261]
Anti-catarrhal activity	*Sinapis alba*, *S. officinale*, *S. erysimoides*, *B. napus*, and *B. nigra*	[39,251,258,260,261,265]
Bronchitis	*S. officinale*, *S. orientale*, and *S. erysimoides*	[259,260,261]
Anti-asthmatic activity	*S. officinale*	[38,39,40,41,258]
Emetic activity	*Sinapis alba*, *B. nigra*, *B. juncea*, *S. officinale*, and *B. nigra*	[123,256,266]
Anti-cancer activity	*B. juncea*, *B. napus*, *B. rapa*, *S. officinale*, and *B compestris*	[39,123,158,256,267]
Effect on bowl	*Sinapis alba*, *B. nigra*, *B. juncea*, and *S. officinale* are used as laxatives. *B. nigra* is used as a carminative	[123,256,257,262,266]
Rubefacient	*B. rapa*, *B. juncea*, and *S. officinale*	[24,30,39,256,268]
Galactagogue	*B. juncea*	[256]
Anti-gout potential	*B. napus*, and *B. rapa*	[21,123,255]
Use in gall stone	*B. napus*, and *B. rapa*	[162,255,265]
Use against alopecia	*B. nigra*	[257]
Anti-dandruff activity	*B. nigra*	[257]
Use in neuralgia	*B. nigra*	[257]
Anti-spasmodic activity	*B. nigra*, and *S. officinale*	[257]
Aphrodisiac activity	*B. rapa*, and *B. nigra*	[24,268]
Use in hepatic and kidney colic	*B. rapa*	[162,255]
Anti-inflammatory activity	*B. rapa*, *S. erysimoides*, and *S. officinale*	[44,255,269]
Anthelmintic activity	*S. orientale*, and *B. rapa* var. *campestris*	[259,270]
Remedial use in fever	*S. orientale*, and *Erysimum repandum*	[127,259]
Use in dysentery	*S. orientale*	[259]
Anti-addiction activity	*S. officinale*	[271]
Use in the disorders voice and throat	*S. officinale*	[40]
Appetising and digestive activities	*S. officinale*, *B. nigra*, and *B. juncea*	[40,41,256,257,258]
Snake bite antidote	*B. rapa* var. *campestris*, and *S. officinale*	[40,41,258,270]
Skin disorders	*Neslia paniculata*	[35,263]

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
