# Peer review of "Brassicaceae Mustards: Phytochemical Constituents, Pharmacological Effects, and Mechanisms of Action against Human Disease"

_ijms, 2024, doi:10.3390/ijms25169039_

Round 1

Reviewer 1 Report

Comments and Suggestions for Authors

The work is very comprehensive but, to complete its content with other good or bad effects of mustard, I recommend for the authors:

- to complete chapter 3 with Antiproliferative activity of the mustard extracts, respectively: antioxidant activity, anti-obesity and anti-hyperglycemia effect;

- To present other effects of mustard: the immune, gastrointestinal, blood, nervous system, reproductive risks, as well as gastrointestinal effects, hematological effects, neurological effects

Author Response

Many thanks to you and all the three reviewers for their very positive gesture and creative comments towards the manuscript. Your pragmatic spotting and addressing them will improve the manuscript for the better.

The comments of the three reviewers and the reply to the reviewer’s comments are presented in the tables in the attachment word file.

Reviewer 2 Report

Comments and Suggestions for Authors

This paper is a review about the chemicals present in Brassicaceae « mustards » and their effect in human health. It includes all the types of phytochemicals present in these plants with the most of the compounds found in seeds, and plenty of the proved and hypothetical effects of these compounds or plant extracts on different type of human diseases. It is a very interesting source of biological effects of Brassicaceae compounds, well written and exhaustive.

However in my opinion the authors need to redefine a bit the scope of their review, beginning with more scientific terms (« mustards » and species, see the following)

Major:

1) In the first line the authors do not define what they call mustards, for it is neither a usual nor a taxonomic name. By the gathering of Brassica napus, I kind of understand that in the mind of the authors, mustards are « Brassicaceae used for their seeds, with a condiment use or an oil use », so it excludes Brassica oleracea used as a legume or Eutremum japonicum used for their roots with a condiment use for instance. But this exclusion is rather artificial and it is the first time I read this classification. Moreover if seeds and their use make the difference between what is called mustards and the other Brassicaceae, then why addressing phytoconstituents of other parts of the plant all along the review?

More consistently in my opinion the authors should focus their reviex on seeds and their constituents only, for all cultivated Brassicaceae. That should highlight the originality of their review when comparing with the several other reviews about health effects of phytochemicals in Brassicaceae (Mann, 2013 ; Jahangir 2009, Shankar 2019, Ramirez, 2020). I suggest that the authors cite these reviews when explaining in their introduction how their own review has an original approach unlike these.

2) Note that Brassica campestris is not the right naming nowadays, but Brassica rapa var. campestris. And note that Brassica alba is not the right naming nowadays, but Sinapis alba.

3) Canola is neither a species nor a scientific name, but it includes varieties of Brassica napus and Brassica rapa with low erucic acid and glucosinolate contents. So please do not use the term Canola or be always accurate about the species.

4) Please add also Camelina sativa, a Brassicaceae species that is cultivated and used for the oil of its seeds

Among other reviews in the same theme are :

Mann, S. K., Khanna, N., & Kaur, S. (2013). Health promoting effects of phytochemicals from brassicaceae: a review. Indian J. Pharm. Biol. Res, 1, 120-131.

Jahangir, M., Kim, H. K., Choi, Y. H., & Verpoorte, R. (2009). Health‐affecting compounds in Brassicaceae. Comprehensive Reviews in Food Science and Food Safety, 8(2), 31-43

Shankar, S., Segaran, G., Sundar, R. D. V., Settu, S., & Sathiavelu, M. (2019). Brassicaceae-A classical review on its pharmacological activities. Int. J. Pharm. Sci. Rev. Res, 55(1), 107-113.

Ramirez, D., Abellán-Victorio, A., Beretta, V., Camargo, A., & Moreno, D. A. (2020). Functional ingredients from Brassicaceae species: Overview and perspectives. International journal of molecular sciences, 21(6), 1998.

Minor:

P7 table 3: keep all units in mg and µg, and get rid of mg/g, mg/100g. get rid of +/- 0.02 too

P9 Table 4:

* please get rid of « Molecular weight… HPLC [142] » in the caption of table 4, because it is wrong

* on the right of beta-D-glucose, no KHSO4 should be there, please erase it.

* please add the raw formula to all glucosinolates as you did for the last glucosinolates of the table, or erase it for all

P14 Table 5:

* this table is not very clear, please separate the glucosinolates and their metabolites at least by carriage return, and turn the line of the nitriles into a line with real names

* Add raw formula to the two last lines

P16 fig 1 : please make the figure more simple by using only two structures, the parent structure for phenolic acids, and the parent structure for cinnamic acids

P18 l79 : mould is a fungus so please mention only fungus as it includes mould

P29 l612: please replace the mistake « cardiac effects » with « allergic effects »  

Author Response

(The authors gave the same response as above.)

Reviewer 3 Report

Comments and Suggestions for Authors

The present paper is a well-documented review-article and recent bibliography has been used, too.

This study aims to explore the relationship between the long-established use of mustards in traditional medicine and the bioactive molecules they contain, while also investigating their potential mechanisms of action. Furthermore, it provides a comprehensive review of the current scientific knowledge regarding the phytochemical composition and biological activities associated with Brassicaceae oilseeds and mustards, offering insights into their potential therapeutic applications. The medicinal properties of these mustard species can be attributed to the diverse array of bioactive compounds they produce, including sulfur-containing glucosinolates, omega-3 fatty acids, terpenoids, phenylpropanoids, flavonoids, tannins, S-methyl cysteine sulfoxide, and trace elements. The diverse components of mustard plants have been utilized for their medicinal properties since ancient times, and their relevance continues to the present day. The wide range of bioactivities and chemical nature of these compounds make them intriguing starting points for the development of novel therapeutics.

In my opinion, this study represent an important work that could be helpful to researchers. The authors provided important information regarding the components of mustard plants and their properties, which can facilitate a deeper understanding of their potential therapeutic effects and contribute to the development of evidence-based dosage regimens that maximize health benefits while ensuring safety. By combining traditional knowledge with contemporary scientific approaches, we can bridge the gap between ancient medicinal practices and modern healthcare solutions

The paper is well organized, easy readable and presented in a well-structured manner.

Therefore, I recommend that the authors address the following aspects to enhance the quality of their study:

1.     Some of the papers cited in this study were not included in the reference list. It is necessary to revise the following papers that have been cited:

Line 48/page 3 - Greve, 1973 #252; Hedrick, 1972 #476

Line 41 / page 8 - Andrew, 2004 #291

Line 43/ page 8 - Wittstock, 2002 #1576

Line 72 / page 18 - Eichel, 2020 #2576;Jo, 2018 #3334; Lietzow, 2021 #3335

Line 135-136/ page 19 - Bajpai, 2023 #3338; Barbosa Pelegrini, 2011 #2015; Jain, 1993 #3336; Rahman, 2020 #3256; Rahman, 2020 #2896

2.     The authors could expand the Conclusions.

3.     The entire reference list have to be formatted according to the journal instructions.

Thank you for the opportunity to review this work! Best regards!

Author Response

(The authors gave the same response as above.)
